# Learning to Emulate Chaos: Adversarial Optimal Transport Regularization

Gabriel Melo [* 1]   Leonardo Santiago [* 2 3]   Peter Y. Lu [4]

## Abstract

Chaos arises in many complex dynamical systems, from weather to power grids, but is difficult to accurately model with data-driven methods such as machine learning emulators. While emulators are promising tools for accelerating simulations and solving inverse problems, they still struggle to learn chaotic dynamics, where sensitivity to initial conditions renders exact long-term forecasts infeasible, especially given noisy data. Recent work instead trains emulators to match the statistical properties of chaotic attractors, but these approaches often rely on handcrafted summary statistics or large, diverse multi-environment datasets. In this work, we propose a family of adversarial optimal transport objectives that can jointly learn high-quality summary statistics and a physically consistent emulator from a single noisy trajectory. We theoretically analyze and experimentally validate a Sinkhorn divergence formulation (2-Wasserstein) and a WGAN-style dual formulation (1-Wasserstein) of our approach. Numerical experiments across a variety of chaotic systems, including ones with high-dimensional spatiotemporal chaos, show that emulators trained using our proposed objectives have significantly improved long-term statistical fidelity.

## 1. Introduction

Chaos is a generic feature of high-dimensional nonlinear dynamical systems (Medio & Lines, 2001) and is fundamental to our understanding of statistical physics (Dorfman, 1999).

Chaotic dynamics are present in many scientific and engineering systems, from turbulent flows (Davidson, 2015) in fluid mechanics, weather and ocean dynamics, and plasma physics to instabilities (Strogatz, 2015) in artificial systems like power grids, traffic flow, and other complex networks. Modeling and simulating chaotic dynamics is therefore critical to solving difficult problems that help address important societal challenges like climate modeling (Lai et al., 2024).

In many real-world applications, the underlying system dynamics are at least partially unknown and need to be learned directly from data. Recently developed data-driven emulators promise improved accuracy and significant speed-ups over traditional modeling approaches, including in domains such as weather prediction (Pathak et al., 2022; Lam et al., 2023; Kochkov et al., 2024; Watt-Meyer et al., 2023), fluid and plasma physics (Mathews et al., 2021), and molecular dynamics (Unke et al., 2021; Musaelian et al., 2023). This is achieved by fitting black box neural network-based surrogate models, such as neural operator architectures (Li et al., 2021; Lu et al., 2021), directly to trajectory data.

However, directly training emulators on noisy chaotic trajectories is difficult due to the defining properties of chaos such as the sensitivity to initial conditions (Medio & Lines, 2001). We show in Section 5.2 that the standard mean squared error (MSE) loss becomes an increasingly poor objective for training emulators on chaotic dynamics over long rollouts, often causing surrogate models to degrade catastrophically during long-term forecasting (Chattopadhyay et al., 2024; Bonavita, 2024). Instead, approaches based on attractor statistics, like the one proposed in this work, provide much needed regularization that encourages the emulator to correctly capture the long-term behavior of chaotic systems.

In this work[1], we use adversarial optimal transport regularization to adaptively learn highly informative summary statistics. The distribution of these statistics is then enforced, via the optimal transport cost, on an emulator that learns the dynamics of the chaotic system. In practice, the emulator $\hat{\mathbf{u}}_{t+1} = g(\mathbf{u}_t)$, which learns the time evolution of the system, and the summary map $f(\mathbf{u})$, which provides the learned summary statistics, are trained simultaneously. The emulator is trained to *minimize* both the standard MSE

---

[*]Equal contribution [1]LTCI, Télécom Paris, Institut Polytechnique de Paris, Palaiseau [2]Department of Mechanical and Aerospace Engineering, North Carolina State University, Raleigh, NC [3]Work performed while at the University of Campinas [4]Department of Electrical and Computer Engineering, Tufts University, Medford, MA. Correspondence to: Gabriel Melo <gabriel.silva@ip-paris.fr>, Leonardo Santiago <lsantia3@ncsu.edu>, Peter Y. Lu <peter.lu@tufts.edu>.

*Proceedings of the 43$^{rd}$ International Conference on Machine Learning*, Seoul, South Korea. PMLR 306, 2026. Copyright 2026 by the author(s).

[1]The code is available at https://github.com/gabrielmelo00/LearningToEmulateChaos_ICML26.

loss and an optimal transport cost that matches the summary statistic distribution of the model to the data, while the summary map is trained to *maximize* the same optimal transport cost. Unlike choosing a set of handcrafted summary statistics that may not be informative enough to constrain the model to a high-dimensional chaotic attractor, this adversarial objective for the summary map ensures it learns an optimally discriminative and therefore highly informative set of summary statistics. This results in an efficient method for ensuring the statistics of the trajectories produced by the emulator match the statistics of the true chaotic attractor.

## 1.1. Contributions

1. **New statistics-based losses for emulating chaos.** We introduce a family of *adversarial optimal transport objectives* that regularize the standard squared error prediction loss for emulator training. These new losses *learn to emulate* chaotic systems by simultaneously learning (i) an adversarial set of informative summary statistics and (ii) an emulator trained to match the distribution of these statistics to the noisy trajectory data. In practice, we use *computationally tractable relaxations* of the $p$-Wasserstein cost in our proposed loss, including a Wasserstein GAN-style loss for $p = 1$ and a Sinkhorn loss that provides an entropy-regularized relaxation valid for any choice of $p \geq 1$.

2. **Theoretical analysis of the proposed adversarial regularization and its behavior in noisy settings.** Our analysis shows that the new loss is *well-behaved* and provides an efficient way to *capture the structure of high-dimensional chaotic attractors*. In particular, given a Lipschitz constraint on the summary map, the new objective is bounded by both the emulator MSE (dynamical) and the Wasserstein distance to the true high-dimensional attractor (statistical). We also show that, while compounding error from noise can significantly degrade direct prediction losses like MSE, our statistics-based loss is much more robust to noise.

3. **Empirical validation of chaotic dynamics learned from a noisy trajectory.** Based on *long-term statistical metrics*, our approach successfully trains emulators to simulate complex chaotic systems using only a single noisy trajectory. In particular, we test on *three high-dimensional chaotic dynamical systems* (Lorenz-96, Kuramoto-Sivashinsky, Kolmogorov flow) and show that our method outperforms standard short-term MSE training as well as prior statistics-based approaches based on handcrafted features.

## 2. Background and Problem Setting

**Notations.** Let $(\mathcal{X}, d_{\mathcal{X}})$ and $(\mathcal{Y}, d_{\mathcal{Y}})$ be metric spaces. We denote by $\mathcal{M}_+(\mathcal{X})$ the set of positive Radon probability

measures on $\mathcal{X}$. Given a measurable map $T : \mathcal{X} \to \mathcal{Y}$ and a measure $\mu \in \mathcal{M}_+(\mathcal{X})$, the pushforward measure $T_\#\mu \in \mathcal{M}_+(\mathcal{Y})$ is defined by $T_\#\mu(B) := \mu(T^{-1}(B))$ for any measurable set $B \subseteq \mathcal{Y}$. The set of joint probability measures on $\mathcal{X} \times \mathcal{Y}$ with marginals $\mu \in \mathcal{M}_+(\mathcal{X})$ and $\nu \in \mathcal{M}_+(\mathcal{Y})$ is defined as $\Pi(\mu, \nu) = \{\pi \in \mathcal{M}_+(\mathcal{X} \times \mathcal{Y}) : \pi(A \times \mathcal{Y}) = \mu(A)$ and $\pi(\mathcal{X} \times B) = \nu(B) \quad \forall A, B \subset \mathcal{X} \times \mathcal{Y}\}$. The Kullback-Leibler divergence between two measures $\mu$ and $\nu$ is $\mathrm{KL}(\mu|\nu)$. We denote by $\mathcal{C}(\mathcal{X}, \mathcal{Y})$ the class of continuous maps $\mathcal{X} \to \mathcal{Y}$. The $\| \cdot \|_F$ is the Frobenius norm.

### 2.1. Dynamical Systems

**Setup.** Let $(\mathcal{U}, d_{\mathcal{U}})$ be a compact metric space (e.g., a compact attractor). The true one-step dynamics is a Borel-measurable map $\Phi : \mathcal{U} \to \mathcal{U}$ admitting an invariant and ergodic probability measure $\mu$, i.e. $\Phi_\#\mu = \mu$. We observe a stationary trajectory $(\mathbf{u}_t)_{t=0}^{T-1}$ generated by

$$\mathbf{u}_{t+1} = \Phi(\mathbf{u}_t), \quad \mathbf{u}_t \sim \mu,$$

so that $(\mathbf{u}_t)_{t=0}^{T-1}$ forms a stationary ergodic Markov chain with marginal law $\mu$.

**Definition 2.1** (Emulator). An emulator is a Borel-measurable map $g : \mathcal{U} \to \mathcal{U}$ used as a one–step predictor of the dynamics $\Phi$. Given a state $\mathbf{u}_t \in \mathcal{U}$, it outputs the prediction $\hat{\mathbf{u}}_{t+1} := g(\mathbf{u}_t) \in \mathcal{U}$. When the input is distributed according to $\mu$, the emulator induces the pushforward measure $\hat{\mu} := g_\#\mu$ on $\mathcal{U}$.

**Definition 2.2** (Rollout). Given an emulator $g : \mathcal{U} \to \mathcal{U}$, we denote by $g^{\circ k}$ its $k$-fold composition. The corresponding $k$-step autonomous rollout from state $\mathbf{u}_t$ is defined as $\hat{\mathbf{u}}_{t+k} := g^{\circ k}(\mathbf{u}_t)$.

**Learning objective and chaotic dynamics.** We are interested in learning an emulator $g$ of the dynamics $\Phi$ in the chaotic regime. In chaotic systems, $\Phi$ exhibits sensitive dependence on initial conditions, so that long-term trajectory prediction is intrinsically unstable. Nevertheless, such systems admit a natural invariant physical measure (Medio & Lines, 2001) supported on the chaotic attractor, which is preserved by the dynamics and governs the long-term statistical behavior. By ergodicity, time averages along a single typical trajectory converge to expectations under this invariant measure. For any observable $f : \mathcal{U} \to \mathbb{R}$,

$$\lim_{T \to \infty} \frac{1}{T} \sum_{t=0}^{T-1} f(\mathbf{u}_t) = \int_{\mathcal{U}} f(\mathbf{u}) \, d\mu(\mathbf{u}) = \mathbb{E}_\mu[f(\mathbf{u})], \quad (1)$$

making $\mu$ the fundamental object that characterizes the asymptotic dynamics. Accordingly, our goal is not to match trajectories, but to learn an emulator whose induced dynamics preserve the invariant statistics of $\Phi$.

## 2.2. Optimal Transport

The optimal transport (OT) cost of transporting between the two probability measures $\mu, \nu$ living on $\mathcal{X}$ is defined as the solution of the Monge-Kantorovich problem

$$\mathcal{W}_c(\mu, \nu) = \min_{\pi \in \Pi(\mu, \nu)} \int_{\mathcal{X} \times \mathcal{X}} c(x, x') \, \mathrm{d}\pi(x, x'). \quad (2)$$

For $c(x, x') = d_{\mathcal{X}}(x, x')^p$ and $p \geq 1$, $\mathcal{W}_c^{1/p}$ is the $p$-Wasserstein distance. The entropic regularized OT optimization (Cuturi, 2013) writes

$$\mathcal{W}_c^{\varepsilon}(\mu, \nu) = \min_{\pi \in \Pi(\mu, \nu)} \int c(x, x') \, \mathrm{d}\pi(x, x') + \varepsilon \, \mathrm{KL}(\pi | \mu \otimes \nu).$$

This formulation is differentiable (Peyré, 2025) and its empirical version is efficiently solved via the Sinkhorn algorithm (Sinkhorn, 1964).

To eliminate the entropic bias (where $\mathcal{W}_c^{\varepsilon}(\mu, \mu) \neq 0$), we employ the Sinkhorn Divergence:

$$S_c^{\varepsilon}(\mu, \nu) := \mathcal{W}_c^{\varepsilon}(\mu, \nu) - \frac{1}{2}\mathcal{W}_c^{\varepsilon}(\mu, \mu) - \frac{1}{2}\mathcal{W}_c^{\varepsilon}(\nu, \nu), \quad (3)$$

ensuring $S_c^{\varepsilon}(\mu, \nu) = 0$ if and only if $\mu = \nu$, interpolating between OT and Maximum Mean Discrepancy (MMD) (Feydy et al., 2019).

## 3. Related Work

**Statistics-based regularization for emulating chaos.** Training emulators on chaotic dynamics using standard squared-error losses is fundamentally limited by the exponential divergence of trajectories. Li et al. (2022) addressed long-term behavior by proposing a Sobolev norm loss. However, this approach struggles in noisy settings where the Sobolev norm becomes dominated by noise.

Several works have approached the problem of preserving invariant measures directly. Yang et al. (2023) used optimal transport to fit parameterized low-dimensional dynamical models, but the PDE-constrained formulation scales poorly. Botvinick-Greenhouse et al. (2023) modeled dynamics through Fokker–Planck equations using Wasserstein distances, but this requires estimating full probability distributions on mesh grids. Platt et al. (2023) proposed constraining reservoir computers by explicitly enforcing dynamical invariants such as the Lyapunov spectrum and fractal dimension. Schiff et al. (2024) proposed DySLIM, which uses Maximum Mean Discrepancy (MMD) regularization to match invariant measures.

Alternative approaches leverage Koopman theory or architecture-specific techniques. Cheng et al. (2025) introduced the Poincaré Flow Neural Network, which uses Koopman theory to linearize chaotic evolution in a learned feature space, addressing contraction and measure invariance without explicit distributional losses. For RNN architectures, Mikhaeil et al. (2022) demonstrated the inherent difficulty of training on chaos, while Hess et al. (2023) proposed generalized teacher forcing for chaotic time series. Jiang et al. (2025) proposed an architecture inspired by implicit integration, and Wang et al. (2024) proved fundamental limitations of closure models for chaotic systems and proposed physics-informed neural operators as an alternative.

Most closely related to our work, Jiang et al. (2023) introduced two approaches for training neural operators to preserve invariant measures. Their first approach uses an optimal transport loss to match distributions of hand-crafted summary statistics, but requires expert knowledge to select informative statistics. Their second approach uses contrastive learning to automatically extract time-invariant features, but requires a diverse multi-environment dataset containing trajectories from many different dynamical regimes.

Our work addresses both limitations by *adversarially learning* summary statistics jointly with the emulator. This approach automatically discovers informative statistics without requiring domain expertise and can learn from a single noisy trajectory.

**Optimal transport and adversarial objectives.** Adversarial objectives provide a flexible framework for distribution matching, most prominently through Generative Adversarial Networks (GANs) (Goodfellow et al., 2014). Wasserstein GANs (WGANs) replace the standard GAN loss with the dual formulation of the 1-Wasserstein distance, leading to improved training stability and more informative gradients (Arjovsky et al., 2017; Gulrajani et al., 2017). In practice, the Lipschitz constraint required by the Kantorovich–Rubinstein dual is only approximately enforced, and parametrized critics do not generally provide a consistent approximation of the true Wasserstein distance (Mallasto et al., 2019; Stanczuk et al., 2021).

Kernel-based discrepancies such as MMD (Gretton et al., 2012) offer stable and unbiased objectives, but do not explicitly capture geometric structure and may struggle when distributions concentrate on low-dimensional manifolds embedded in high-dimensional spaces (Bińkowski et al., 2018). Optimal transport metrics naturally address these settings by handling non-overlapping supports and encoding geometry, but their direct use in learning is hindered by computational cost, lack of smoothness, and statistical instability. Entropic regularization mitigates these issues by yielding smooth, efficiently computable OT objectives via Sinkhorn iterations (Cuturi, 2013). Following Genevay et al. (2018), by parameterizing the ground cost and optimizing it to distinguish between model and data distributions, this framework becomes an adversarial min-max game. Here, a neural network

learns a feature-space geometry that adaptively maximizes the discrepancy, forcing the generator to match the data distribution across its most salient geometric features.

# 4. Our Approach: Adversarial Optimal Transport Regularization

Motivated by the fact that chaotic dynamics are characterized by their invariant (physical) measure rather than long-term trajectory accuracy, we adapt adversarial optimal transport ideas to the dynamical setting. Specifically, we compare the pushforward measures induced by the true dynamics $\Phi$, with invariant distribution $\mu$, and by its emulator $g$, with induced distribution $\hat{\mu} := g_{\#}\mu$. Rather than learning a discriminator or a ground cost in data space, we introduce an adversarially learned summary map that exposes discrepancies in invariant statistics. This leads to a min–max objective directly aligned with preserving the physical measure of the system.

Let $(\mathcal{S}, d_{\mathcal{S}})$ denote the *summary space*, and let $\mathcal{F} \subset \mathcal{C}(\mathcal{U}, \mathcal{S})$ be a family of summary maps $f : \mathcal{U} \to \mathcal{S}$, with $\mathbf{s} = f(\mathbf{u})$. When $\mathbf{u} \sim \mu$, the induced distribution of summaries is given by $\mathbf{s} \sim f_{\#}\mu$. Fixing the optimal transport cost to be $c = d_{\mathcal{S}}{}^p$ for some $p \in [1, \infty)$, we consider the adversarial objective

$$\min_{g \in \mathcal{G}} \max_{f \in \mathcal{F}} \mathcal{L}(g, f) :=$$
$$= \mathcal{L}_{\mathrm{MSE}}\big(\mathbf{u}_{t+1}, g(\mathbf{u}_t)\big) + \lambda \mathcal{W}_{d_{\mathcal{S}}{}^p}\big(f_{\#}\mu, f_{\#}\hat{\mu}\big), \quad (4)$$

in which $\lambda > 0$. The MSE term enforces local one-step fidelity, while the Wasserstein term targets agreement of invariant statistics in the summary space.

The maximization over $\mathcal{F}$ encourages the model to search for summary statistics that highlight discrepancies between the true and emulator-induced invariant measures. By minimizing this worst-case discrepancy, the emulator $g$ is driven to match the long-term statistical behavior of the data, so that no summary within the chosen family can reliably distinguish the model from the true dynamics.

The Wasserstein term in Eq. (4) is non-differentiable with respect to both $g$ and $f$ and is defined at the population level, rendering direct optimization via stochastic gradient methods intractable. In practice, this issue is addressed by replacing the exact Wasserstein distance with differentiable and sample-efficient relaxations, such as entropic regularization or dual formulations, which admit stochastic optimization (Montesuma et al., 2024).

Figure 1 summarizes the proposed framework, where an emulator is trained with a one-step prediction loss regularized by optimal transport between real and generated trajectory distributions via a learnable summary map. We next describe WGAN-style and Sinkhorn-based instantiations.

## 4.1. WGAN-style dual formulation

When $p = 1$, the Wasserstein term in Eq. (4) can be reformulated using the Kantorovich–Rubinstein duality. Specifically, the 1-Wasserstein distance between the pushforward measures $f_{\#}\mu$ and $f_{\#}\hat{\mu}$ can be written

$$\mathcal{W}_{d_{\mathcal{S}}{}^1}(f_{\#}\mu, f_{\#}\hat{\mu}) =$$
$$= \sup_{\varphi \in \mathrm{Lip}_1(\mathcal{S})} \left( \mathbb{E}_{\mathbf{s} \sim f_{\#}\mu}[\varphi(\mathbf{s})] - \mathbb{E}_{\hat{\mathbf{s}} \sim f_{\#}\hat{\mu}}[\varphi(\hat{\mathbf{s}})] \right) \quad (5)$$

where $\mathrm{Lip}_1(\mathcal{S})$ denotes the set of 1-Lipschitz real-valued functions $\varphi : \mathcal{S} \to \mathbb{R}$, commonly referred to as critics. Substituting (5) into the objective (4), and retaining the adversarial maximization over summaries $f \in \mathcal{F}$, yields

$$\min_{g \in \mathcal{G}} \max_{f \in \mathcal{F}} \max_{\varphi \in \mathrm{Lip}_1(\mathcal{S})} \mathcal{L}_{\mathrm{W}}(g, f) := \quad (6)$$
$$\mathcal{L}_{\mathrm{MSE}}\big(\mathbf{u}_{t+1}, g(\mathbf{u}_t)\big) + \lambda \Big( \mathbb{E}_\mu[\varphi \circ f(\mathbf{u})] - \mathbb{E}_\mu[\varphi \circ f \circ g(\mathbf{u})] \Big).$$

This formulation corresponds to a WGAN-style objective in the summary space, where the critic $\varphi$ acts as a statistical witness distinguishing the pushforward invariant measures. Unlike classical WGANs operating on i.i.d. samples, the expectations here are taken with respect to the invariant measure $\mu$ induced by the underlying dynamics. The resulting objective is differentiable almost everywhere in all parameters and can be optimized using stochastic gradient methods with standard Lipschitz-enforcing heuristics, such as gradient penalization (Gulrajani et al., 2017), weight clipping (Arjovsky et al., 2017) and spectral normalization of the weights (Miyato et al., 2018).

## 4.2. Sinkhorn divergence

WGAN-style methods rely on the dual formulation of the 1-Wasserstein distance and require explicit enforcement of Lipschitz constraints on the critic. We instead adopt a primal, entropically regularized optimal transport objective based on the Sinkhorn divergence, which provides stable, fully differentiable gradients without requiring explicit Lipschitz constraints on a critic (Cuturi, 2013; Genevay et al., 2018).

Following the learnable cost framework of Genevay et al. (2018), we define a feature map $f : \mathcal{U} \to \mathcal{S}$ and induce a ground cost

$$c_f(\mathbf{u}, \hat{\mathbf{u}}) := d_{\mathcal{S}}{}^p\big(f(\mathbf{u}), f(\hat{\mathbf{u}})\big), \quad (7)$$

where $d_{\mathcal{S}}$ is a fixed metric on the feature space $\mathcal{S}$ (e.g., Euclidean) and $p \geq 1$. This construction measures discrepancies after embedding states into a learned summary space.

The resulting Sinkhorn divergence $S_{c_f}^{\varepsilon}(\mu, \hat{\mu})$ is equivalent to computing the Sinkhorn divergence between the pushforward measures $f_{\#}\mu$ and $f_{\#}\hat{\mu}$ with respect to the fixed

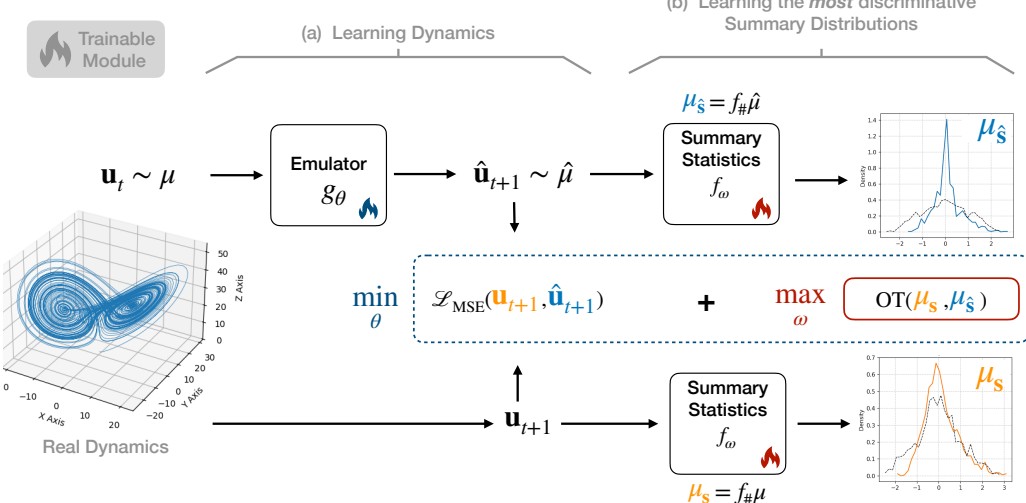

*Figure 1.* Adversarial optimal transport regularization for emulating chaotic dynamics. (a) Emulator training via one-step prediction loss with OT regularization. (b) Adversarial learning of summary statistics that maximize the discrepancy between real and generated trajectory distributions while minimizing the full loss.

ground metric $d_{\mathcal{S}}{}^p$. Replacing the Wasserstein term in Eq. (4) yields the objective

$$\min_{g \in \mathcal{G}} \max_{f \in \mathcal{F}} \mathcal{L}_S(g, f) := \mathcal{L}_{\text{MSE}}\big(\mathbf{u}_{t+1}, g(\mathbf{u}_t)\big) + \lambda\, S_{c_f}^{\varepsilon}\big(\mu, \hat{\mu}\big),$$
(8)

where optimizing over $f$ learns a task-adaptive geometry that emphasizes relevant distributional discrepancies.

## 5. Theoretical Analysis

### 5.1. Properties of the proposed objectives

In this section, we study the properties of the losses introduced above. We first establish a bound showing that the optimal transport regularizer (in Eq.4) is controlled by the one-step prediction error. By Proposition 5.1, the OT term is uniformly bounded in Eq.4. Moreover, for $p = 2$, it is controlled by the mean-squared prediction error, implying that near an MSE minimizer the OT regularizer remains bounded and cannot dominate the objective. This provides a local stability guarantee for the combined loss, but does not by itself ensure tractability or existence of a saddle point for the full min–max problem. Corollary 5.2 extends this control to multi-step rollouts, linking long-horizon distributional discrepancies to rollout accuracy.

**Proposition 5.1** (Dynamical summary space Wasserstein bound). *Given the true one-step dynamics* $\Phi : \mathcal{U} \to \mathcal{U}$ *with invariant measure* $\mu$, *and the emulator* $g : \mathcal{U} \to \mathcal{U}$ *with induced measure* $\hat{\mu} = g_\# \mu$. *Let* $f : \mathcal{U} \to \mathcal{S}$ *be an* $L_f$-*Lipschitz summary map, and fix* $p \in [1, \infty)$. *Then,*

$$\mathcal{W}_{d_{\mathcal{S}}{}^p}\big(f_\# \mu, f_\# \hat{\mu}\big) \le L_f{}^p\, \mathbb{E}_{\mathbf{u}_t \sim \mu}[d_{\mathcal{U}}\big(\mathbf{u}_{t+1}, g(\mathbf{u}_t)\big)^p].$$

**Corollary 5.2** ($k$-step rollout Wasserstein bound). *Under the assumptions of Proposition 5.1, let* $\Phi^{\circ k}$ *and* $g^{\circ k}$ *denote*

the $k$–fold compositions of the true dynamics $\Phi$ and emulator $g$, respectively, and define the $k$–step emulator measure $\hat{\mu}_k := (g^{\circ k})_\# \mu$. Then, for any $k \in \mathbb{N}$ and any $p \in [1, \infty)$,

$$\mathcal{W}_{d_{\mathcal{S}}{}^p}\big(f_\# \mu, f_\# \hat{\mu}_k\big) \le L_f{}^p\, \mathbb{E}_{\mathbf{u}_t \sim \mu}\Big[d_{\mathcal{U}}\big(\Phi^{\circ k}(\mathbf{u}_t), g^{\circ k}(\mathbf{u}_t)\big)^p\Big].$$

Then, Proposition 5.3 provides an upper bound on the adversarial summary-space objective in terms of the $p$-Wasserstein distance defined on the original state space $\mathcal{U}$. Corollary 5.4 further shows that, when the summary hypothesis class is sufficiently expressive to recover the state-space geometry up to a global scaling, this bound is tight and the adversarial objective exactly reduces to the state-space Wasserstein distance.

**Proposition 5.3** (General summary space Wasserstein bound). *Let* $f : \mathcal{U} \to \mathcal{S}$ *be an* $L_f$-*Lipschitz summary map, and fix* $p \in [1, \infty)$. *Then,*

$$\mathcal{W}_{d_{\mathcal{S}}{}^p}(f_\# \mu, f_\# \hat{\mu}) \le L_f{}^p\, \mathcal{W}_{d_{\mathcal{U}}{}^p}(\mu, \hat{\mu}).$$

**Corollary 5.4** (General reduction to state space Wasserstein objective). *If* $\mathcal{U} \subseteq \mathcal{S} \subseteq \mathbb{R}^n$, $d_{\mathcal{S}} = d_{\mathcal{U}}$ *are the Euclidean metric on* $\mathbb{R}^n$. *Let* $\mathcal{F} \subseteq \text{Lip}_{L_f}(\mathcal{U})$ *be a class of summary maps that contains the scaling map* $f(\mathbf{u}) = L_f \mathbf{u}$. *Then,*

$$\sup_{f \in \mathcal{F}} \mathcal{W}_{d_{\mathcal{S}}{}^p}(f_\# \mu, f_\# \hat{\mu}) = L_f{}^p\, \mathcal{W}_{d_{\mathcal{U}}{}^p}(\mu, \hat{\mu}).$$

*Remark* 5.5. The results above highlight a central requirement for learning task-adaptive summary spaces via optimal transport: the hypothesis class of summary maps must be suitably controlled. Without structural constraints on $f$, the adversarial objective may become ill-posed, lose stability, or fail to admit meaningful guarantees. In this work, we

enforce this through a global Lipschitz constraint on the summary hypothesis class, which guarantees stability and bounds the summary-space transport cost by state-space prediction errors. Other forms of control, such as bounded or spectral constraints, could provide similar guarantees.

## 5.2. Robustness to noisy trajectory data

In practice, observed trajectories are corrupted by measurement noise, and emulator predictions are initialized from noisy estimates of the system state. We now analyze how such perturbations affect the MSE and Wasserstein components of the objective (4), revealing a fundamental advantage of distributional regularization for chaotic systems.

**Noise model.** We consider the following sources of Gaussian noise:

(i) *Measurement noise:* $\mathbf{u}_{t+1}^{\mathrm{obs}} := \Phi(\mathbf{u}_t) + \boldsymbol{\epsilon}_1$ where $\boldsymbol{\epsilon}_1 \sim \mathcal{N}(0, \sigma_1^2 I_d)$.

(ii) *Initial condition noise:* $\hat{\mathbf{u}}_{t+1}^{\mathrm{noisy}} := g(\mathbf{u}_t + \boldsymbol{\epsilon}_2)$ where $\boldsymbol{\epsilon}_2 \sim \mathcal{N}(0, \sigma_2^2 I_d)$.

Both noise terms are assumed independent of each other and of the state $\mathbf{u}_t \sim \mu$. In most cases, $\sigma_1^2 = \sigma_2^2$ since both types of noise originate from a noisy training trajectory.

**Definition 5.6** (Noisy measures). Let $\mathcal{N}_\sigma := \mathcal{N}(0, \sigma^2 I_d)$ denote the isotropic Gaussian on $\mathbb{R}^d$. Define:

- The *observed measure*: $\mu^{\mathrm{obs}} := \mu * \mathcal{N}_{\sigma_1}$ (convolution),
- The *noisy initial measure*: $\tilde{\mu} := \mu * \mathcal{N}_{\sigma_2}$,
- The *noisy emulator measure*: $\hat{\mu}^{\mathrm{noisy}} := g_{\#}\tilde{\mu}$.

### 5.2.1. MSE UNDER NOISE

The noisy MSE loss takes the form

$$\mathcal{L}_{\mathrm{MSE}}^{\mathrm{noisy}} := \mathbb{E}_{\mathbf{u}_t \sim \mu, \, \boldsymbol{\epsilon}_1, \, \boldsymbol{\epsilon}_2}\left[\left\| \Phi(\mathbf{u}_t) + \boldsymbol{\epsilon}_1 - g(\mathbf{u}_t + \boldsymbol{\epsilon}_2) \right\|^2\right]. \quad (9)$$

**Proposition 5.7** (MSE noise decomposition). *Under the noise model above, the noisy MSE decomposes as*

$$\mathcal{L}_{\mathrm{MSE}}^{\mathrm{noisy}} = \mathcal{L}_{\mathrm{MSE}}^{\mathrm{clean}} + d\sigma_1^2 + \sigma_2^2 \, \mathbb{E}_{\mathbf{u}_t \sim \mu}\big[\|Dg(\mathbf{u}_t)\|_F^2\big] + O(\sigma_2^4),$$

*where $\mathcal{L}_{\mathrm{MSE}}^{\mathrm{clean}} := \mathbb{E}_{\mathbf{u}_t \sim \mu}[\|\Phi(\mathbf{u}_t) - g(\mathbf{u}_t)\|^2]$ is the noise-free MSE, $Dg$ denotes the Jacobian of $g$.*

*Remark* 5.8. For chaotic systems, the Jacobian norm $\|Dg(\mathbf{u}_t)\|_F$ grows exponentially under iteration.

**Corollary 5.9** (MSE blow-up for chaotic systems). *Let $\lambda_{max} > 0$ denote the maximal Lyapunov exponent of the emulator dynamics $g$. For $k$-step rollouts with noisy initial conditions, the MSE satisfies*

$$\mathcal{L}_{\mathrm{MSE}}^{(k),\mathrm{noisy}} \gtrsim d\sigma_1^2 + \sigma_2^2 \cdot d \cdot e^{2\lambda_{max}k} \quad \text{as } k \to \infty. \quad (10)$$

*In particular, even for a perfect emulator ($g = \Phi$), the noisy MSE diverges exponentially in the rollout horizon $k$.*

*Remark* 5.10. Corollary 5.9 reveals a fundamental limitation of trajectory-based losses for chaotic systems: the MSE objective becomes dominated by initial condition noise at long horizons, regardless of emulator quality. This motivates the use of distributional objectives that are insensitive to trajectory-level divergence.

### 5.2.2. WASSERSTEIN UNDER NOISE

We now analyze the Wasserstein component of the objective (4) under noise. Throughout this section, we work with the $p$-Wasserstein distance

$$W_p(\mu, \nu) := \mathcal{W}_{d_{\mathcal{U}^p}}(\mu, \nu)^{1/p}, \quad (11)$$

which satisfies the triangle inequality, rather than the cost $\mathcal{W}_{d_{\mathcal{U}^p}}$ directly.

**Proposition 5.11** (Wasserstein noise bounds). *Let $\boldsymbol{\epsilon} \sim \mathcal{N}(0, \sigma^2 I_d)$ and define the Gaussian moment constant*

$$\kappa_{p,d} := \mathbb{E}\big[\|Z\|^p\big]^{1/p}, \quad Z \sim \mathcal{N}(0, I_d), \quad (12)$$

*which satisfies $\kappa_{p,d} = \Theta(\sqrt{d})$ for fixed $p$ as $d \to \infty$. For any $p \in [1, \infty)$:*

(i) Measurement noise: $W_p(\mu, \mu^{\mathrm{obs}}) \leq \sigma_1 \kappa_{p,d}$.

(ii) Initial condition noise: $W_p(\mu, \tilde{\mu}) \leq \sigma_2 \kappa_{p,d}$.

**Definition 5.12** (Exponential mixing). An emulator $g : \mathcal{U} \to \mathcal{U}$ with invariant measure $\nu$ is *exponentially mixing in $W_p$* with rate $\rho \in (0, 1)$ if there exists $C > 0$ such that for any probability measure $\eta$ on $\mathcal{U}$ with finite $p$-th moment,

$$W_p\big((g^{\circ k})_{\#}\eta, \nu\big) \leq C\rho^k \quad \text{for all } k \in \mathbb{N}. \quad (13)$$

*Remark* 5.13. On the compact metric space $(\mathcal{U}, d_{\mathcal{U}})$, exponential mixing in $W_p$ for any $p \in [1, \infty)$ implies exponential mixing in $W_q$ for all $q \in [1, \infty)$, with rate depending on $p$, $q$, and $\mathrm{diam}(\mathcal{U})$. Thus the choice of $p$ in Definition 5.12 is largely a matter of convenience.

**Theorem 5.14** (Wasserstein noise forgetting). *Suppose $g$ is exponentially mixing in $W_p$ with invariant measure $\nu$, rate $\rho \in (0, 1)$, and constant $C > 0$. Let $\hat{\mu}_k^{\mathrm{noisy}} := (g^{\circ k})_{\#}\tilde{\mu}$ denote the $k$-step noisy emulator measure. Then*

$$W_p\big(\mu^{\mathrm{obs}}, \hat{\mu}_k^{\mathrm{noisy}}\big) \leq$$
$$\underbrace{\sigma_1 \kappa_{p,d}}_{\text{measurement noise}} + \underbrace{W_p(\mu, \nu)}_{\text{model error}} + \underbrace{C\rho^k}_{\text{IC noise (vanishing)}}. \quad (14)$$

*In particular,*

$$\lim_{k \to \infty} W_p\big(\mu^{\mathrm{obs}}, \hat{\mu}_k^{\mathrm{noisy}}\big) \leq \sigma_1 \kappa_{p,d} + W_p(\mu, \nu). \quad (15)$$

*Remark* 5.15 (Physical interpretation). Theorem 5.14 captures a fundamental property of chaotic dynamics: initial condition information is forgotten in the distributional sense. While individual trajectories diverge exponentially (causing MSE blow-up per Corollary 5.9), the statistical ensemble converges to the invariant measure regardless of initial perturbations. The Wasserstein distance inherits this insensitivity, making it well-suited for long-horizon evaluation of chaotic emulators.

The following corollary extends the noise forgetting result to the adversarial summary space objective in (4).

**Corollary 5.16** (summary space Wasserstein noise forgetting). *Under the assumptions of Theorem 5.14, let $f : \mathcal{U} \to \mathcal{S}$ be an $L_f$-Lipschitz summary map. Then*

$$W_p\big(f_\# \mu^{\mathrm{obs}}, f_\# \hat{\mu}_k^{\mathrm{noisy}}\big) \leq L_f\Big(\sigma_1 \kappa_{p,d} + W_p(\mu, \nu) + C\rho^k\Big),$$

*and therefore,*

$$\lim_{k \to \infty} W_p\big(f_\# \mu^{\mathrm{obs}}, f_\# \hat{\mu}_k^{\mathrm{noisy}}\big) \leq L_f\Big(\sigma_1 \kappa_{p,d} + W_p(\mu, \nu)\Big),$$

5.2.3. COMPARISON AND IMPLICATIONS

The key observation is that the MSE loss becomes dominated by initial condition noise at long horizons, regardless of emulator quality, while the Wasserstein distance stabilizes to a finite value determined only by measurement noise and model error in preserving the invariant measure.

This analysis suggests a natural horizon-dependent training strategy. Define the mixing time $\tau_{\mathrm{mix}} := -1/\log \rho$, and consider the following scenarios.

- *Short horizon* ($k \ll \tau_{\mathrm{mix}}$): Both MSE and Wasserstein objectives are sensitive to IC noise, but MSE provides meaningful trajectory-level gradients.

- *Long horizon* ($k \gg \tau_{\mathrm{mix}}$): The MSE becomes noise-dominated and uninformative. Only the Wasserstein term provides a useful signal toward the invariant measure.

This motivates the combined objective with horizon-dependent weighting:

$$\mathcal{L}(g, f) := \sum_{k=1}^{K} w_k \mathcal{L}_{\mathrm{MSE}}^{(k)} + \lambda \mathcal{W}_{d_\mathcal{S}^p}(f_\# \mu, f_\# \hat{\mu}), \quad (16)$$

where setting $w_k \to 0$ for $k \gg \tau_{\mathrm{mix}}$ down-weights noise-dominated MSE terms, while the Wasserstein regularizer ensures long-term statistical consistency even when trajectory-level prediction is fundamentally impossible due to chaos. In practice, we conservatively choose $w_1 = 1$, $w_{k>1} = 0$.

*Remark* 5.17 (Scope of noise model). The additive isotropic Gaussian noise model is adopted for analytical tractability.

The two key phenomena, exponential MSE blow-up (Corollary 5.9) and Wasserstein noise forgetting (Theorem 5.14), are not artifacts of this choice. MSE blow-up follows from the exponential amplification of any initial condition perturbation through the system Jacobian, a defining property of chaotic dynamics (Remark 5.8). Wasserstein robustness follows from exponential mixing (Definition 5.12), which renders the precise form of the initial perturbation irrelevant over long horizons. Measurement noise of a different distributional form would similarly perturb the observed measure without disrupting the attractor geometry. Alternative noise models would affect the explicit constants in the bounds derived below, but not the qualitative behavior.

## 6. Numerical Experiments

We evaluate our adversarial OT regularization framework on three benchmark chaotic systems of increasing spatial complexity. Across all experiments, we compare our two learnable OT regularizers, Sinkhorn and WGAN (Section 4), against the an MSE-only baseline (No OT) and the handcrafted OT baseline (Fixed OT) of Jiang et al. (2023). All methods share the same emulator architecture; they differ in the OT regularization term and its associated training procedure.

**Dynamical systems.** The *Lorenz-96* (L96) system with 60 variables is a canonical high-dimensional chaotic model for atmospheric dynamics. The *Kuramoto-Sivashinsky* (KS) equation discretized on 256 grid points, is a canonical one-dimensional chaotic PDE modeling reaction-diffusion instabilities and flame front propagation (Hyman & Nicolaenko, 1986). *Kolmogorov flow* is a two-dimensional incompressible Navier-Stokes system driven by a sinusoidal body force, widely used as a benchmark for turbulent dynamics (Chandler & Kerswell, 2013). We additionally include *Lorenz-63* (L63) as a low-dimensional test, where we restrict the analysis to qualitative visualizations of attractor geometry. Implementation details for all four systems are deferred to Appendices D.1, D.2, D.3 and D.4, respectively.

**Training regimes.** For L96, we consider two regimes. The *multi-trajectory* regime follows Jiang et al. (2023) exactly, using 2,000 trajectories of length 2,000 with forcing $F^{(n)} \sim \mathrm{Unif}[10, 18]$, allowing a direct comparison against their reported numbers under identical data and architecture conditions. The *single-trajectory* regime uses a single trajectory of the same length at fixed $F = 10$, and is the setting used for all KS and Kolmogorov experiments, reflecting the scenario where only one observed run is available. The data is corrupted with additive Gaussian noise with standard deviation $\sigma \cdot \mathrm{std}$, where the scaling factor $\sigma$ is the noise level and $\mathrm{std}$ denotes the empirical standard deviation of the corresponding clean trajectory.

*Table 1.* Summary statistics used to compute the $L^1$ histogram error at evaluation time.

| System | Fixed Summary Map $S(\mathbf{u})$ |
|--------|-----------------------------------|
| L96 | $\left\{\partial_t \mathbf{u}_i,\ (\mathbf{u}_{i+1} - \mathbf{u}_{i-2})\mathbf{u}_{i-1},\ \mathbf{u}_i\right\}$ |
| KS | $\left\{\partial_t \mathbf{u},\ \partial_x \mathbf{u},\ \partial_{xx}\mathbf{u}\right\}$ |
| Kolmogorov Flow | $\left\{\mathbf{v} \cdot \nabla\omega,\ \partial_t\omega,\ \omega\right\}$ |

**Evaluation protocol.** Models trained on noisy data ($\sigma = 0.3$) are evaluated in two settings: *clean rollouts* ($\sigma = 0$), consisting of autoregressive predictions initialized from noisy states and rolled out on 200 noise-free test trajectories for $1{,}500$ steps (L96) or $1{,}000$ steps (KS, Kolmogorov); and *noisy rollouts* ($\sigma = 0.3$), evaluating on trajectories corrupted at the same noise level as training. We report three metrics: one-step **RMSE** (short-term accuracy), **spectral distance** (energy spectrum fidelity), and $L^1$ **histogram error** (invariant measure alignment). We denote by $\mathbf{u}$ the primary state variable of each system: $\mathbf{u}$ is the velocity field for L96 and KS, and $\mathbf{u} = \omega = (\nabla \times \mathbf{v}) \cdot \hat{z}$ is the vorticity for Kolmogorov flow. The histogram error is always computed using fixed and system-specific summary statistics given by $S(\mathbf{u})$ listed in Table 1, regardless of the summary representation used during training, ensuring a fair comparison across fixed and learnable methods. Full metric definitions are given in Appendix B.

*Remark* 6.1. For chaotic systems, the metrics of primary interest are the $L^1$ histogram error and spectral distance, as they directly measure alignment with the invariant measure of the attractor. RMSE captures only short-term trajectory accuracy and, as shown in Section 5.2, degrades exponentially with rollout horizon due to sensitivity to initial conditions. We include it for completeness.

**Architecture choices.** For L96 and KS, the emulator is a Fourier Neural Operator (FNO; Li et al. 2021), following Jiang et al. (2023). For Kolmogorov flow, we use a UNet encoder-decoder with spectral convolution layers, following Jiang et al. (2025). The learnable summary map $f : \mathcal{U} \to \mathbb{R}^{\dim(\mathcal{U}) \times d}$ shares the same output structure as the fixed one $S$ (Table 1), which correspond to the special case $d = 3$ with handcrafted features. By default, $f$ is a three-layer MLP with hidden width 128 for all systems, except where we additionally test a spatially aware variant: a 1D convolutional map for KS and a 2D convolutional map with circular padding for Kolmogorov flow, both providing inductive biases appropriate to the periodic structure of their respective domains.

## 7. Discussion

Our approaches (Sinkhorn, WGAN) for learning informative summary statistics consistently outperform the standard MSE-only baseline (No OT) and match or outperform OT

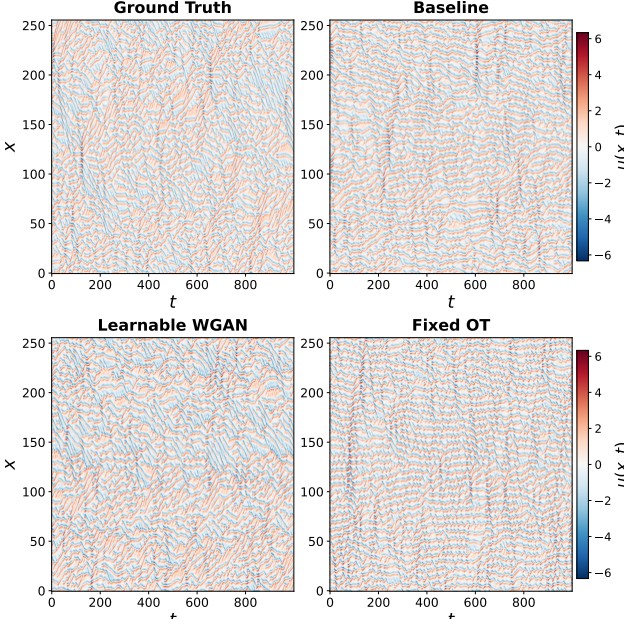

*Figure 2.* KS full roll-out evaluation (clean data). Our WGAN-style emulator most faithfully replicates the diagonal wave patterns and spatial structure of the ground truth (numerical simulation) across the full evaluation rollout.

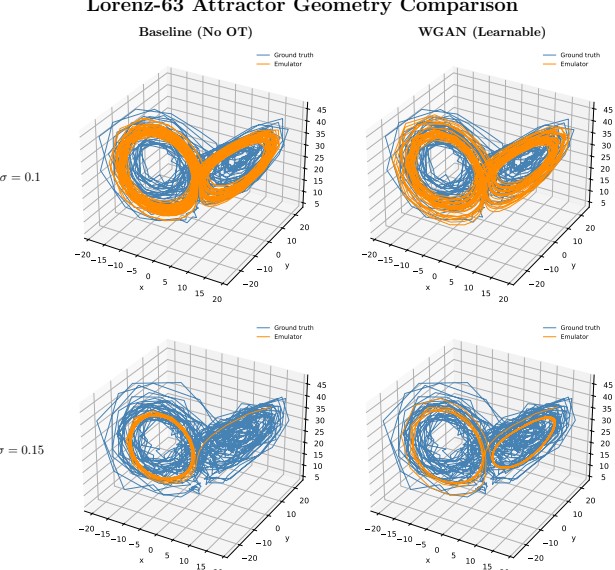

*Figure 3.* L63 emulator geometry at increasing noise level $\sigma$. At $\sigma = 0.10$, the MSE baseline (No OT) underestimates the spatial extent of the attractor; at $\sigma = 0.15$, it collapses to a limit cycle, losing the bilobal structure entirely. WGAN maintains coverage of both lobes at both noise levels, directly illustrating how distributional regularization prevents attractor collapse under noise. More results on L63 are provided in Appendix F.

regularization with handcrafted summary statistics (Fixed OT) on key statistical quantities across all evaluated systems (Table 2). All OT-regularized methods outperform the No OT baseline, confirming the benefit of distributional regularization for long-term statistical fidelity. Fixed OT performs strongly on $L^1$ histogram error, as expected from

*Table 2.* Experimental results across all benchmark systems. Each system block reports median performance under noisy ($\sigma = 0.3$) and clean ($\sigma = 0.0$) conditions. Best result per metric per block is **bolded** and our methods are highlighted with a shaded background. Unless otherwise stated, $d = 3$.

| System | Method | Noisy ($\sigma = 0.3$) | | | Clean ($\sigma = 0.0$) | | |
|---|---|---|---|---|---|---|---|
| | | $L^1$ Hist.↓ | Spec. dist.↓ | RMSE↓ | $L^1$ Hist.↓ | Spec. dist.↓ | RMSE↓ |
| L96 *(multi-traj)* | No OT (baseline) | 0.348 | 0.321 | **0.388** | 0.225 | 0.294 | **0.302** |
| | Fixed OT | 0.175 | 0.141 | 0.398 | **0.055** | 0.119 | 0.313 |
| | Sinkhorn (ours) | 0.175 | 0.112 | 0.406 | 0.109 | 0.083 | 0.325 |
| | WGAN (ours) | **0.149** | **0.049** | 0.401 | 0.083 | **0.074** | 0.318 |
| | WGAN ($d=1$) (ours) | 0.234 | 0.205 | 0.392 | 0.091 | 0.176 | 0.306 |
| L96 *(single-traj)* | No OT (baseline) | 0.298 | 0.307 | **0.362** | 0.187 | 0.321 | **0.271** |
| | Fixed OT | 0.178 | 0.151 | 0.368 | **0.077** | 0.157 | 0.278 |
| | Sinkhorn (ours) | 0.222 | **0.137** | 0.367 | 0.082 | **0.123** | 0.278 |
| | WGAN (ours) | **0.151** | 0.145 | 0.371 | 0.115 | 0.175 | 0.283 |
| KS *(single-traj)* | No OT (baseline) | 0.454 | 0.351 | **0.370** | 0.241 | 0.342 | **0.243** |
| | Fixed OT | 0.290 | 0.349 | 0.379 | 0.310 | 0.386 | 0.257 |
| | Sinkhorn (ours) | 0.435 | 0.219 | 0.373 | 0.190 | 0.212 | 0.246 |
| | WGAN (ours) | 0.339 | **0.148** | 0.375 | **0.153** | **0.183** | 0.251 |
| | WGAN (1D-Conv) (ours) | **0.259** | 0.198 | 0.393 | 0.245 | 0.205 | 0.278 |
| Kolmogorov Flow *(single-traj)* | No OT ((baseline) | 0.660 | 0.249 | 0.381 | 0.266 | 0.205 | **0.202** |
| | Fixed OT | 0.614 | 0.168 | **0.381** | 0.239 | 0.193 | 0.203 |
| | Sinkhorn (2D-Conv) (ours) | 0.360 | **0.144** | 0.405 | **0.178** | **0.129** | 0.221 |
| | WGAN (2D-Conv) (ours) | **0.273** | 0.157 | 0.424 | 0.190 | 0.139 | 0.225 |

a method explicitly trained to match those statistics, yet our learnable OT methods match or outperform it in the noisy evaluation while remaining competitive in the clean evaluation. The clearest advantage of learnable summaries appears in spectral distance, a metric not enforced by the Fixed OT objective, where our learnable OT methods consistently outperform both baselines across all systems. This demonstrates the broad coverage achieved by the summary statistics learned via adversarial OT regularization.

*Table 3.* Leading Lyapunov exponent of trained models on single trajectory L96. Implementation details on LLE estimation are provided in Appendix B.1.

| | Ground Truth | No OT (baseline) | Fixed OT | Sinkhorn (ours) | WGAN (ours) |
|---|---|---|---|---|---|
| **LLE** | 2.334 | 1.615 | 1.890 | 2.030 | **2.336** |

Beyond attractor statistics, our approach also improves the dynamical properties of the emulator, such as the leading Lyapunov exponent (LLE), which represents the rate at which chaos becomes unpredictable (Table 3). For Lorenz-96, our WGAN-style approach produces an emulator with an LLE of 2.336, nearly matching the ground truth value of 2.334, while the No OT baseline and Fixed OT fall significantly short. Note that Koopman-based methods (Cheng et al., 2025) are structurally precluded from reproducing positive Lyapunov exponents, as any stable finite-dimensional linear system must have LLE $\leq 0$, whereas chaos requires LLE $> 0$ by definition.

Qualitatively, we consistently observe that the learned emulators produce more realistic chaotic rollouts (Figure 2). For the low-dimensional Lorenz-63 system, we can directly visualize the effect of OT regularization on the geometry of the attractor (Figure 3), illustrating how our approaches improve the shape of the learned attractor and prevent collapse in high-noise settings. Additional visualizations are provided in Appendix G.

## 8. Conclusion

We have proposed a family of adversarial optimal transport objectives for training emulators of chaotic dynamical systems. Without requiring any prior knowledge of the system, this approach jointly learns high-quality summary statistics and a physically consistent emulator from a single chaotic trajectory. Our results demonstrate the effectiveness of this type of learnable regularizer in enforcing the statistics of complex, high-dimensional chaotic attractors. While this approach is currently restricted to ergodic systems with a well-defined attractor, we anticipate that our method can be extended to transient dynamics or mildly time-dependent systems by adjusting the time range over which statistics are computed. Future work will also explore extending this framework to stochastic systems, where capturing the interplay between stochasticity and chaotic dynamics presents a natural and challenging generalization.

## Acknowledgements

This research was supported by the French National Research Agency (ANR) through the PEPR IA FOUNDRY project (ANR-23-PEIA-0003). This work is supported by the National Science Foundation under Cooperative Agreement PHY-2019786 (The NSF AI Institute for Artificial Intelligence and Fundamental Interactions, http://iaifi.org/).

## Impact Statement

This paper presents work whose goal is to advance the field of machine learning. There are many potential societal consequences of our work, none of which we feel must be specifically highlighted here.

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

# A. Theoretical Results

## A.1. Proofs of Section 5.1

### A.1.1. PROPOSITION 5.1

*Proof of Proposition 5.1.* Since $\Phi_{\#}\mu = \mu$, we have the identity of pushforwards

$$f_{\#}\mu = f_{\#}(\Phi_{\#}\mu) = (f \circ \Phi)_{\#}\mu, \qquad f_{\#}\hat{\mu} = f_{\#}(g_{\#}\mu) = (f \circ g)_{\#}\mu.$$

Thus

$$\mathcal{W}_{d_{\mathcal{S}^p}}(f_{\#}\mu, f_{\#}\hat{\mu}) = \mathcal{W}_{d_{\mathcal{S}^p}}((f \circ \Phi)_{\#}\mu, (f \circ g)_{\#}\mu).$$

Consider the coupling $\pi := (\Phi \times g)_{\#}\mu$ on $\mathcal{U} \times \mathcal{U}$, i.e., sample $\mathbf{u} \sim \mu$ and pair $\Phi(\mathbf{u})$ with $g(\mathbf{u})$. Its first and second marginals are $\Phi_{\#}\mu = \mu$ and $g_{\#}\mu = \hat{\mu}$, so $\pi \in \Pi(\mu, \hat{\mu})$. Pushing $\pi$ forward by $(f, f)$ gives a feasible coupling

$$\tilde{\pi} := (f, f)_{\#}\pi \in \Pi((f \circ \Phi)_{\#}\mu, (f \circ g)_{\#}\mu).$$

By definition of the $p$-Wasserstein objective,

$$\mathcal{W}_{d_{\mathcal{S}^p}}((f \circ \Phi)_{\#}\mu, (f \circ g)_{\#}\mu) \leq \int d_{\mathcal{S}}(f(\Phi(\mathbf{u})), f(g(\mathbf{u})))^p \, \mathrm{d}\mu(\mathbf{u}).$$

Using that $f$ is $L_f$-Lipschitz, $d_{\mathcal{S}}(f(\Phi(\mathbf{u})), f(g(\mathbf{u}))) \leq L_f \, d_{\mathcal{U}}(\Phi(\mathbf{u}), g(\mathbf{u}))$,

$$\int d_{\mathcal{S}}(f(\Phi(\mathbf{u})), f(g(\mathbf{u})))^p \, \mathrm{d}\mu(\mathbf{u}) \leq L_f{}^p \int d_{\mathcal{U}}(\Phi(\mathbf{u}), g(\mathbf{u}))^p \, \mathrm{d}\mu(\mathbf{u}).$$

Finally, writing $\mathbf{u}_t \sim \mu$ and $\mathbf{u}_{t+1} = \Phi(\mathbf{u}_t)$ leads to the desired equation. $\square$

### A.1.2. COROLLARY 5.2

*Proof of Corollary 5.2.* Since $\mu$ is invariant for $\Phi$, we have $(\Phi^{\circ k})_{\#}\mu = \mu$ for all $k \in \mathbb{N}$. Applying Proposition 5.1 with $\Phi$ replaced by $\Phi^{\circ k}$ and $g$ replaced by $g^{\circ k}$ yields

$$\mathcal{W}_{d_{\mathcal{S}^p}}(f_{\#}\mu, f_{\#}\hat{\mu}_k) = \mathcal{W}_{d_{\mathcal{S}^p}}((f \circ \Phi^{\circ k})_{\#}\mu, (f \circ g^{\circ k})_{\#}\mu) \leq L_f{}^p \, \mathbb{E}_{\mathbf{u}_t \sim \mu} \left[ d_{\mathcal{U}}(\Phi^{\circ k}(\mathbf{u}_t), g^{\circ k}(\mathbf{u}_t))^p \right].$$

$\square$

### A.1.3. PROPOSITION 5.3

*Proof of Proposition 5.3.* Let $\pi^* \in \Pi(\mu, \hat{\mu})$ be the optimal $p$-Wasserstein coupling on $\mathcal{U} \times \mathcal{U}$. Pushing $\pi^*$ forward by $(f, f)$ gives a feasible coupling $(f, f)_{\#}\pi^* \in \Pi(f_{\#}\mu, f_{\#}\hat{\mu})$ on $\mathcal{S} \times \mathcal{S}$. By definition of the $p$-Wasserstein objective and then using that $f$ is $L_f$-Lipschitz,

$$\mathcal{W}_{d_{\mathcal{S}^p}}(f_{\#}\mu, f_{\#}\hat{\mu}) \leq \int d_{\mathcal{S}}(f(\mathbf{u}), f(\hat{\mathbf{u}}))^p \, \mathrm{d}\pi^*(\mathbf{u}, \hat{\mathbf{u}})$$

$$\leq L_f{}^p \int d_{\mathcal{U}}(\mathbf{u}, \hat{\mathbf{u}})^p \, \mathrm{d}\pi^*(\mathbf{u}, \hat{\mathbf{u}})$$

$$= L_f{}^p \, \mathcal{W}_{d_{\mathcal{U}^p}}(\mu, \hat{\mu}).$$

$\square$

### A.1.4. COROLLARY 5.4

*Proof of Corollary 5.4.* By Proposition 5.3, for any $f \in \mathcal{F} \subseteq \mathrm{Lip}_{L_f}(\mathcal{U})$,

$$\mathcal{W}_{d_{\mathcal{S}^p}}(f_{\#}\mu, f_{\#}\hat{\mu}) \leq L_f{}^p \, \mathcal{W}_{d_{\mathcal{U}^p}}(\mu, \hat{\mu}). \tag{17}$$

Taking the supremum over $f \in \mathcal{F}$ yields the upper bound.

To show that this bound is tight, consider the linear map $f(\mathbf{u}) = A\mathbf{u}$ with $A = L_f I$, whose operator norm satisfies

$$\|A\|_{\mathrm{op}} := \sup_{\mathbf{v} \neq 0} \frac{\|A\mathbf{v}\|_2}{\|\mathbf{v}\|_2} = L_f.$$

Since $f(\mathbf{u}) = L_f \mathbf{u}$ is $L_f$-Lipschitz with respect to the Euclidean metric, it belongs to the closure of $\mathcal{F}$ by assumption. Combining this with (17) proves the claim. $\square$

## A.2. Proofs of Section 5.2

### A.2.1. PROPOSITION 5.7

*Proof of Proposition 5.7.* Expanding the squared norm in (9):

$$\mathcal{L}_{\mathrm{MSE}}^{\mathrm{noisy}} = \mathbb{E}\Big[\big\|\Phi(\mathbf{u}_t) - g(\mathbf{u}_t + \boldsymbol{\epsilon}_2)\big\|^2\Big] + \mathbb{E}\big[\|\boldsymbol{\epsilon}_1\|^2\big] + 2\,\mathbb{E}\Big[\big\langle \Phi(\mathbf{u}_t) - g(\mathbf{u}_t + \boldsymbol{\epsilon}_2),\, \boldsymbol{\epsilon}_1 \big\rangle\Big].$$

The cross term vanishes by independence of $\boldsymbol{\epsilon}_1$, and $\mathbb{E}[\|\boldsymbol{\epsilon}_1\|^2] = d\sigma_1^2$.

For the first term, Taylor-expand $g(\mathbf{u}_t + \boldsymbol{\epsilon}_2)$ around $\boldsymbol{\epsilon}_2 = 0$:

$$g(\mathbf{u}_t + \boldsymbol{\epsilon}_2) = g(\mathbf{u}_t) + Dg(\mathbf{u}_t)\,\boldsymbol{\epsilon}_2 + O(\|\boldsymbol{\epsilon}_2\|^2).$$

Substituting and using $\mathbb{E}[\boldsymbol{\epsilon}_2] = 0$:

$$\mathbb{E}\Big[\big\|\Phi(\mathbf{u}_t) - g(\mathbf{u}_t + \boldsymbol{\epsilon}_2)\big\|^2\Big] = \mathbb{E}\Big[\big\|\Phi(\mathbf{u}_t) - g(\mathbf{u}_t) - Dg(\mathbf{u}_t)\boldsymbol{\epsilon}_2\big\|^2\Big] + O(\sigma_2^4)$$
$$= \mathcal{L}_{\mathrm{MSE}}^{\mathrm{clean}} + \mathbb{E}\big[\|Dg(\mathbf{u}_t)\boldsymbol{\epsilon}_2\|^2\big] + O(\sigma_2^4),$$

where the cross term again vanishes. Finally, $\mathbb{E}[\|Dg(\mathbf{u}_t)\boldsymbol{\epsilon}_2\|^2] = \sigma_2^2\,\mathbb{E}[\|Dg(\mathbf{u}_t)\|_F^2]$. $\square$

### A.2.2. COROLLARY 5.9

*Proof of Corollary 5.9.* For the $k$-step rollout, Proposition 5.7 applies with $g$ replaced by $g^{\circ k}$. By the chain rule, $D(g^{\circ k})(\mathbf{u}) = \prod_{j=0}^{k-1} Dg(g^{\circ j}(\mathbf{u}))$. For a chaotic system with maximal Lyapunov exponent $\lambda_{\max}$, the leading singular value grows as $\|D(g^{\circ k})\|_{\mathrm{op}} \sim e^{\lambda_{\max} k}$, implying $\|D(g^{\circ k})\|_F^2 \gtrsim d \cdot e^{2\lambda_{\max} k}$. $\square$

### A.2.3. PROPOSITION 5.11

*Proof of Proposition 5.11.* For (i) and (ii), let $\sigma \in \{\sigma_1, \sigma_2\}$ and consider the coupling $\pi = (\mathrm{Id}, \mathrm{Id} + \boldsymbol{\epsilon})_{\#}\mu$, where $\boldsymbol{\epsilon} \sim \mathcal{N}(0, \sigma^2 I_d)$ is sampled independently of $\mathbf{u} \sim \mu$. This coupling has marginals $\mu$ and $\mu * \mathcal{N}_\sigma$, so

$$W_p(\mu, \mu * \mathcal{N}_\sigma)^p = \mathcal{W}_{d_{\mathcal{U}}^p}(\mu, \mu * \mathcal{N}_\sigma) \leq \mathbb{E}_{\mathbf{u},\boldsymbol{\epsilon}}\big[\|\boldsymbol{\epsilon}\|^p\big] = \sigma^p\,\mathbb{E}\big[\|Z\|^p\big] = \sigma^p \kappa_{p,d}^p.$$

$\square$

### A.2.4. THEOREM 5.14

*Proof of Thorem 5.14.* By the triangle inequality for $W_p$:

$$W_p\big(\mu^{\mathrm{obs}}, \hat{\mu}_k^{\mathrm{noisy}}\big) \leq W_p(\mu^{\mathrm{obs}}, \mu) + W_p(\mu, \nu) + W_p\big(\nu, (g^{\circ k})_{\#}\tilde{\mu}\big).$$

By Proposition 5.11 (i), the first term is bounded by $\sigma_1 \kappa_{p,d}$. By exponential mixing (Definition 5.12) applied to the initial measure $\tilde{\mu}$, the third term satisfies $W_p(\nu, (g^{\circ k})_{\#}\tilde{\mu}) \leq C\rho^k$. The limit follows immediately. $\square$

### A.2.5. COROLLARY 5.16

*Proof of Corollary 5.16.* Since $f$ is $L_f$-Lipschitz, $W_p(f_{\#}\mu, f_{\#}\nu) \leq L_f\,W_p(\mu, \nu)$ for any measures $\mu, \nu$ on $\mathcal{U}$. The result follows by applying this contraction to (14). $\square$

### A.3. Linear Summary Case

Finally, Proposition A.1 discusses properties of linear summary maps, with an exact result that only exactly holds when the optimal transport map $T^*$ commutes with the summary map $f(\mathbf{u}) = A\mathbf{u}$. Nevertheless, this provides some intuition for the optimal directions chosen by linear summary maps.

**Proposition A.1** (Optimal linear summary via displacement covariance). *Let $\mathcal{U} \subseteq \mathbb{R}^d$ and $\mathcal{S} = \mathbb{R}^k$ with $d_{\mathcal{U}}$, $d_{\mathcal{S}}$ the Euclidean metrics. Suppose $\mu, \hat{\mu}$ are absolutely continuous measures on $\mathcal{U}$ with finite second moments, and let $T^*$ be the Brenier map satisfying $T^*_{\#}\mu = \hat{\mu}$. Define the* displacement covariance

$$C_T := \mathbb{E}_{\mathbf{u} \sim \mu}\big[(\mathbf{u} - T^*(\mathbf{u}))(\mathbf{u} - T^*(\mathbf{u}))^{\top}\big], \tag{18}$$

*with eigenvalues $\sigma_1(C_T) \geq \cdots \geq \sigma_d(C_T) \geq 0$ and eigenvectors $\mathbf{v}_1, \ldots, \mathbf{v}_d$. Then $\mathrm{tr}(C_T) = \mathcal{W}_{d_{\mathcal{U}}^2}(\mu, \hat{\mu})$, and for $\mathcal{F}_k = \{f(\mathbf{u}) = A\mathbf{u} : A \in \mathbb{R}^{k \times d}, \|A\|_{\mathrm{op}} \leq L_f\}$:*

$$\max_{f \in \mathcal{F}_k} \mathcal{W}_{d_{\mathcal{S}}^2}(f_{\#}\mu, f_{\#}\hat{\mu}) \leq L_f^2 \sum_{i=1}^{k} \sigma_i(C_T), \tag{19}$$

*with equality when the Brenier map $T^*$ is separable in the eigenbasis of $C_T$. If equality holds, the optimal linear summary is $A^* = L_f[\mathbf{v}_1 \cdots \mathbf{v}_k]^{\top}$.*

*Proof of Proposition A.1.* By Brenier's theorem, the optimal coupling between $\mu$ and $\hat{\mu}$ is $\gamma^* = (\mathrm{Id}, T^*)_{\#}\mu$, so

$$\mathcal{W}_{d_{\mathcal{U}}^2}(\mu, \hat{\mu}) = \mathbb{E}_{\mathbf{u} \sim \mu}\big[\|\mathbf{u} - T^*(\mathbf{u})\|^2\big] = \mathrm{tr}(C_T).$$

For any linear map $f(\mathbf{u}) = A\mathbf{u}$, the pushforward coupling $(A, A)_{\#}\gamma^*$ is feasible for $(A_{\#}\mu, A_{\#}\hat{\mu})$, hence

$$\mathcal{W}_{d_{\mathcal{S}}^2}(A_{\#}\mu, A_{\#}\hat{\mu}) \leq \mathbb{E}_{\mathbf{u} \sim \mu}\big[\|A(\mathbf{u} - T^*(\mathbf{u}))\|^2\big] = \mathrm{tr}(AC_TA^{\top}). \tag{20}$$

Write the SVD of $A$ as $A = USV^{\top}$ where $U \in \mathbb{R}^{k \times k}$ and $V \in \mathbb{R}^{d \times k}$ have orthonormal columns, and $S = \mathrm{diag}(s_1, \ldots, s_k)$ with $s_1 \geq \cdots \geq s_k \geq 0$. The constraint $\|A\|_{\mathrm{op}} \leq L_f$ implies $s_i \leq L_f$ for all $i$. Then

$$\mathrm{tr}(AC_TA^{\top}) = \mathrm{tr}(V^{\top}C_TVS^2) = \sum_{i=1}^{k} s_i^2 (V^{\top}C_TV)_{ii}.$$

Since $C_T$ has eigenvalues $\sigma_1(C_T) \geq \cdots \geq \sigma_d(C_T)$, the diagonal entries of $V^{\top}C_TV$ for any $V$ with orthonormal columns satisfy $(V^{\top}C_TV)_{ii} \leq \sigma_i(C_T)$ by the Cauchy interlacing theorem, with equality when the columns of $V$ are the eigenvectors $\mathbf{v}_1, \ldots, \mathbf{v}_k$. Subject to $s_i \leq L_f$, the maximum of $\sum_{i=1}^{k} s_i^2 \sigma_i(C_T)$ is achieved by $s_i = L_f$ for all $i$, giving

$$\max_{\|A\|_{\mathrm{op}} \leq L_f} \mathrm{tr}(AC_TA^{\top}) = L_f^2 \sum_{i=1}^{k} \sigma_i(C_T),$$

achieved by $A^* = L_f[\mathbf{v}_1 \cdots \mathbf{v}_k]^{\top}$.

Equality holds in (20) if and only if $(A, A)_{\#}\gamma^*$ is optimal for $(A_{\#}\mu, A_{\#}\hat{\mu})$, i.e., $AT^*(\mathbf{u}) = T_A^*(A\mathbf{u})$ where $T_A^*$ is the Brenier map for the projected measures. For $A^* = L_f[\mathbf{v}_1 \cdots \mathbf{v}_k]^{\top}$, this holds when $T^*$ is separable in the eigenbasis: $\mathbf{v}_i^{\top}T^*(\mathbf{u}) = h_i(\mathbf{v}_i^{\top}\mathbf{u})$ for monotone $h_i$. In this case, the projected transport decouples into $k$ independent 1D optimal transports, and $(A^*, A^*)_{\#}\gamma^*$ is optimal. $\square$

## B. Metrics

Following Jiang et al. (2023), we adopt the same metrics reported in their work.

**RMSE.** We report the one-step *relative* root mean squared error (RMSE) to measure short-term prediction accuracy, defined as

$$\text{RMSE} = \frac{\|\mathbf{u}_{t+1} - \hat{g}_\theta(\mathbf{u}_t,)\|_2}{\|\mathbf{u}_{t+1}\|_2},$$

where $\hat{g}_\theta$ denotes the learned emulator.

**Energy spectrum error.** We compute the relative mean absolute error of the energy spectrum, defined as the squared magnitude of the spatial Fourier transform. Let $\mathcal{F}[\mathbf{u}_t]$ denote the spatial FFT of $\mathbf{u}_t$. The spectral error is computed as

$$\frac{1}{T} \sum_{t \in 1:T} \frac{\left\||\mathcal{F}[\mathbf{u}_t]|^2 - |\mathcal{F}[\hat{\mathbf{u}}_t]|^2\right\|_1}{\left\||\mathcal{F}[\mathbf{u}_t]|^2\right\|_1},$$

where the average is taken over time along the rollout.

$L^1$ **histogram error.** The histogram error is computed as the $L^1$ distance between empirical histograms of summary statistics evaluated on the true and predicted trajectories. For consistency across methods, histograms are always computed using the system-dependent fixed summary presented in Table 1 regardless of whether the model was trained with fixed or learnable summaries.

### B.1. Leading Lyapunov Exponent

The leading Lyapunov exponent (LLE) quantifies the average rate of exponential divergence between nearby trajectories, and serves as a defining signature of chaos. We estimate it using the Benettin method (Benettin et al., 1980). Starting from an initial state $\mathbf{u}_0$, two trajectories are evolved in parallel: a reference $\mathbf{u}_t$ and a perturbed copy $\tilde{\mathbf{u}}_t$, with initial separation $\|\tilde{\mathbf{u}}_0 - \mathbf{u}_0\| = d_0 = 10^{-2}$.

At each rescaling event, triggered whenever the separation $d_t = \|\tilde{\mathbf{u}}_t - \mathbf{u}_t\|$ leaves the band $[10^{-5}, 10]$, we record the local expansion factor $a = d_t/d_0$, rescale the perturbation back to magnitude $d_0$, and accumulate $\log a$. The LLE is then estimated as

$$\lambda = \frac{1}{T} \sum_k \log a_k,$$

where the sum runs over all rescaling events within a window of $T = 1000$ steps, preceded by a warm-up phase of 100 steps to allow the trajectory to settle onto the attractor. The integration step size is $\delta t = 0.1$.

This procedure is applied twice with identical hyperparameters: once using the numerical integrator as the step function to obtain the **ground-truth LLE**, and once using the learned emulator to obtain the **predicted LLE**. Comparing these two values directly quantifies how faithfully the emulator reproduces the chaotic regime of the underlying dynamics.

## C. Lipschitz Regularity of the Summary Map

### C.1. Implicit regularity from joint training

Although no explicit Lipschitz constraint is imposed on $f$ during training, the joint MSE-OT objective implicitly induces regularity. As established in Proposition 5.7 and Corollary 5.9, the MSE term bounds and stabilizes the adversarial OT objective, while the OT term corrects long-term behavior not captured by the MSE loss. Combined with the warm-start strategy (pretraining on MSE before activating the OT term) and early stopping of the maximization phase for the Sinkhorn variant, this confines $f$ to a well-behaved Lipschitz regime throughout optimization.

Table 4 reports the distribution of Jacobian spectral norms $\|Df(x)\|_2$ over the L96 validation set. For WGAN and Sinkhorn with early stopping, norms are strictly positive and well-distributed across the attractor, confirming nontrivial local geometry. Without early stopping, Sinkhorn exhibits degenerate behavior: the majority of norms collapse near zero with rare large spikes, indicating instability in the learned geometry. Early stopping eliminates this entirely.

*Table 4.* Distribution of Jacobian spectral norms $\|Df(x)\|_2$ over the L96 validation set.

| Method | Min | 05th Percentile | Median | 95th Percentile | Max |
|---|---|---|---|---|---|
| **Sinkhorn** | | | | | |
| Early stopping | 0.250 | 0.278 | 0.400 | 0.623 | 0.716 |
| No early stopping | 0.000 | 0.000 | 0.000 | 0.394 | 5.762 |
| **WGAN** | | | | | |
| | 1.290 | 1.366 | 1.638 | 2.009 | 2.126 |

Consistent behavior was observed across train, validation, and test splits, confirming that the learned Lipschitz behavior generalizes and is not an artifact of overfitting.

Let $f(x)$ be an $M$-layer MLP with weight matrices $W_1, \ldots, W_M$ and pointwise activations $\sigma_\ell$ that are 1-Lipschitz (e.g. ReLU). Following (Khromov & Pal Singh, 2024), the global Lipschitz constant of $f$ satisfies

$$\textbf{Upper Bound: } \operatorname{Lip}(f) \leq \prod_{\ell=1}^{M} \|W_\ell\|_2, \tag{21}$$

while the lower bound is given by

$$\textbf{Lower Bound: } \sup_{x \in \mathcal{D}} \|Df(x)\|_2 \leq \operatorname{Lip}(f), \tag{22}$$

where $\mathcal{D}$ is some subset of the data (e.g. train split) and $D$ is the Jacobian. Table 5 reports both bounds evaluated on the L96 training split. In all cases the gap between upper and lower bound remains moderate.

*Table 5.* Empirical Lipschitz bounds on training data L96.

| Method | Upper Bound | Lower Bound |
|---|---|---|
| WGAN | 8.70 | 2.11 |
| Sinkhorn (early stopping) | 1.94 | 0.77 |

This provides practical justification for the $L$-Lipschitz assumption underlying the theoretical analysis.

### C.2. Joint hinge regularization

For settings where explicit control over the Lipschitz constant of $f$ is required, we propose a joint hinge regularization that enforces $L_{\min} \leq \operatorname{Lip}(f) \leq L_{\max}$ during training, turning the theoretical constants into controllable hyperparameters. The regularizer takes the form

$$\mathcal{R}(f) = S_\beta\big(\operatorname{Lip}_{\text{upper}}(f) - L_{\max}\big) + S_\beta\big(L_{\min} - \operatorname{Lip}_{\text{lower}}(f)\big), \tag{23}$$

where $S_\beta(x) = \frac{1}{\beta} \log(1 + e^{\beta x})$ is a smooth approximation to $\max(x, 0)$, recovering the standard hinge as $\beta \to \infty$. Inspired by (Khromov & Pal Singh, 2024) results, we propose

$$\operatorname{Lip}_{\text{upper}}(f) = \prod_{\ell=1}^{M} \|W_\ell\|_2, \tag{24}$$

This penalizes whenever $\text{Lip}_{\text{upper}}(f)$ exceeds $L_{\max}$, preventing the summary map from becoming geometrically irregular. The lower bound term uses

$$\text{Lip}_{\text{lower}}(f) = \frac{1}{|\mathcal{B}|} \sum_{x \in \mathcal{B}} \|Df(x)\|_2, \tag{25}$$

the mean Jacobian spectral norm over a small subset $\mathcal{B}$ of states sampled from the current batch, as a local sensitivity estimate. This penalizes whenever $f$ becomes locally flat, preventing feature collapse. While the maximum Jacobian norm better approximates the true Lipschitz constant (Khromov & Pal Singh, 2024), the mean is more informative for detecting practical collapse, where the map degenerates on most of the attractor rather than at isolated points.

### C.3. Experiments

We validate the regularizer on L96 with the WGAN variant, fixing all architecture and training configuration and varying only $L_{\max}$ (with $L_{\min} = 0.01$ fixed throughout). Table 6 reports the resulting Lipschitz bounds; Table 7 reports performance on noisy and clean trajectories.

*Table 6.* Empirical Lipschitz bounds on training data (WGAN, L96).

| Method | Upper Bound | Lower Bound |
|---|---|---|
| $L_{\max} = 4$ | 3.11 | 0.91 |
| $L_{\max} = 10$ | 7.05 | 1.52 |
| No regularization | 8.69 | 1.61 |

*Table 7.* Effect of Lipschitz regularization on WGAN performance for L96 in the noisy ($\sigma = 0.3$) and clean ($\sigma = 0.0$) settings.

| Method | Noisy ($\sigma = 0.3$) | | | Clean ($\sigma = 0.0$) | | |
|---|---|---|---|---|---|---|
| | L1 hist. $\downarrow$ | Spec. dist. $\downarrow$ | RMSE $\downarrow$ | L1 hist. $\downarrow$ | Spec. dist. $\downarrow$ | RMSE $\downarrow$ |
| $L_{\max} = 4$ | 0.176 | 0.133 | 0.368 | 0.090 | 0.144 | 0.279 |
| $L_{\max} = 10$ | 0.177 | 0.135 | 0.367 | 0.082 | 0.151 | 0.278 |
| No regularization | 0.151 | 0.145 | 0.371 | 0.115 | 0.175 | 0.283 |

The upper bound remains below the prescribed threshold throughout optimization for regularized runs, while the unregularized model naturally operates within a compatible regime (Table 6). Performance is preserved or marginally improved under regularization across both noisy and clean settings. Figure 4 show that bounds remain within the prescribed regime throughout optimization.

**Practical trade-off.** Explicit regularization via per-step Jacobian computation increases training cost by approximately $4\times$. Given that the unregularized model already operates within a stable Lipschitz regime in our setting, we present the unregularized formulation as the primary approach and the regularized variant as a principled option when explicit control over the Lipschitz constant is required.

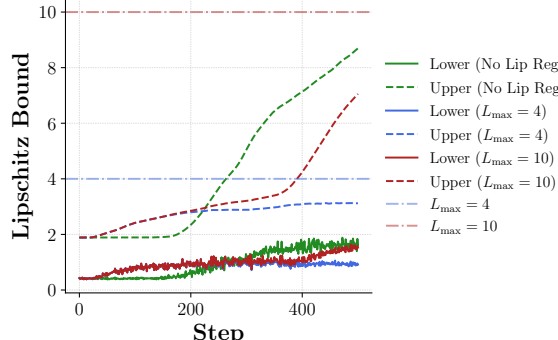

*Figure 4.* Lipschitz bounds during training for the WGAN summary map $f$ on L96, under three regularization settings. Upper bounds (dashed) are computed as $\prod_\ell \|W_\ell\|_2$; lower bounds (solid) are estimated from the mean Jacobian spectral norm over a batch subset. Prescribed thresholds $L_{\max} = 4$ and $L_{\max} = 10$ are shown as horizontal dash-dotted lines.

# D. Experimental Details

## D.1. Experimental details for L-96

**Dataset generation.** We adopt exactly the dataset generation procedure of Jiang et al. (2023). All experiments are conducted on the Lorenz–96 system with 60 variables, defined by

$$\dot{\mathbf{u}}_i = (\mathbf{u}_{i+1} - \mathbf{u}_{i-2})\mathbf{u}_{i-1} - \mathbf{u}_i + F, \tag{26}$$

with cyclic indexing. We generate 2,000 trajectories of length 2,000, where the forcing parameter is sampled independently for each trajectory as $F^{(n)} \sim \text{Unif}[10, 18]$. Observations are corrupted with additive Gaussian noise with standard deviation $0.3 \cdot \text{std}$, where $\text{std}$ denotes the empirical standard deviation of the corresponding clean trajectory.

**Training protocol.** Training follows the same procedure as Jiang et al. (2023). We use teacher forcing over sliding windows of length 100 with stride 2: the emulator predicts odd timesteps while even timesteps are taken from ground truth. The OT weight $\lambda$ is kept the same for the three set-ups ($\lambda = 3$).

**Emulator architecture.** All experiments use the same Fourier Neural Operator (FNO; Li et al. 2021) architecture as in Jiang et al. (2023). The emulator architecture and all associated hyperparameters are kept fixed across all baselines and learnable OT variants, ensuring that differences in performance arise solely from the OT regularization.

**Learnable summary network.** The summary network $f_\theta : \mathbb{R}^m \to \mathbb{R}^{m \times d}$ maps a state $\mathbf{u}_t \in \mathbb{R}^m$ to $m$ embeddings in $\mathbb{R}^d$, one per spatial location. This matches the output structure of the fixed statistics $S$, which also assigns a vector in $\mathbb{R}^d$ to each spatial point, with $d = 3$ hardwired physical quantities (e.g., $\partial_t u_i$, advection, $u_i$ for L96). In both cases, applying $f$ (or $S$) across all timesteps of a trajectory yields a point cloud in $\mathbb{R}^d$, over which the OT metrics between predicted and true trajectories is computed.

**Example D.1.** Take $\mathbf{u}_t = (u_1^t, u_2^t) \in \mathbb{R}^2$ and trajectory $[\mathbf{u}_1, \ldots, \mathbf{u}_T]$. The fixed handcrafted summary map is:

$$\mathbf{u}_t \mapsto \left[ (\partial_t u_1^t, \ \text{adv}_1^t, \ u_1^t) \ (\partial_t u_2^t, \ \text{adv}_2^t, \ u_2^t) \right] \in \mathbb{R}^{2 \times 3}.$$

Our learned MLP summary map is:

$$\mathbf{u}_t \mapsto \text{MLP}(u_1^t, u_2^t) \in \mathbb{R}^{2 \times d}.$$

In both cases, across the full trajectory you collect $T \times 2$ vectors in $\mathbb{R}^d$ as a point cloud.

**Overview of OT regularization strategies.** We compare the fixed-statistics OT baseline introduced by Jiang et al. (2023) against two learnable OT-based regularizers. All three approaches use an OT discrepancy between empirical distributions induced by the emulator, but differ in whether the summary representation is fixed or learned, and whether the OT term is optimized adversarially.

**Fixed OT baseline.** The fixed baseline of Jiang et al. (2023) computes the Sinkhorn divergence between empirical distributions of hand-crafted physics-inspired summaries (c.f. evaluation protocol in Section 6)

$$S(\mathbf{u}_i) = \{\dot{\mathbf{u}}_i, \ (\mathbf{u}_{i+1} - \mathbf{u}_{i-2})\mathbf{u}_{i-1}, \ \mathbf{u}_i\}. \tag{27}$$

The Sinkhorn divergence is computed with entropic regularization $\varepsilon = 0.02$ and $p = 2$. This method is non-adversarial: the summary representation is fixed, and no maximization is performed over the summary space.

**Learnable Sinkhorn regularization.** In the learnable Sinkhorn variant, the fixed summary $S$ is replaced by a learnable summary network $f_\theta$. The resulting objective takes the form of an adversarial min–max game (Eq. (8)), where the emulator minimizes prediction error regularized by a Sinkhorn divergence, while the summary network maximizes this divergence by shaping the summary space. The Sinkhorn divergence uses the same parameters as in the fixed baseline ($\varepsilon = 0.02$, $p = 2$) and is implemented using GeomLoss [2]. Notably, we employed early-stopping (at 350/400 out of 500 training steps). for a more stable training.

---

[2] https://www.kernel-operations.io/geomloss/

**Learnable WGAN regularization.** The learnable WGAN variant estimates the OT discrepancy via adversarial training following Eq. (6). In addition to the summary network $f_\theta$, a critic $\varphi_\phi : \mathbb{R}^d \to \mathbb{R}$ is optimized to maximize the dual formulation of the 1-Wasserstein distance. The emulator minimizes prediction error regularized by this adversarial Wasserstein estimate. Lipschitz continuity of the critic is enforced via weight clipping to the interval $[-0.01, 0.01]$.

Inspired by the results of the theoretical section (Prop 5.1), we use warm-start pretraining on the MSE objective, and subsequent activation of the OT term.

**Summary and critic architectures.** The learnable summary network $f_\theta$ is implemented as a three-layer multilayer perceptron with two hidden layers of width 128 and ReLU activations. For the WGAN variant, the critic $\varphi_\phi$ is implemented as a two-layer multilayer perceptron with hidden dimension 64 and ReLU activations. All architectural choices are kept fixed across experiments. Motivated by the theoretical formulation, we also enforce Lipschitz continuity of the learnable summary network. In both learnable setups (Sinkhorn and WGAN), this is achieved via weight clipping of the summary network parameters.

### D.2. Experimental details for KS

**System and dataset.** We consider the Kuramoto–Sivashinsky equation,

$$\partial_t \mathbf{u} + \mathbf{u}\partial_x\mathbf{u} + \partial_{xx}\mathbf{u} + \partial_{xxxx}\mathbf{u} = 0, \tag{28}$$

discretized on 256 grid points with periodic boundary conditions, under single training trajectory is used with additive Gaussian observation noise with standard deviation $0.3 \cdot \mathrm{std}$, where $\mathrm{std}$ denotes the empirical standard deviation of the corresponding clean trajectory. All models are evaluated over rollouts of 1,000 steps.

**Emulator architecture.** Similarly to the L96 experiments, the same Fourier Neural Operator (FNO; Li et al. 2021) architecture as in Jiang et al. (2023) was used, with an increased capacity demand due to the increased physical dimension.

**Summary and critic architectures.** The MLP summary network follows the same architecture as in the L96 experiments. For the 1D convolutional variant, the MLP is replaced by a stack of 1D convolutions with circular padding, providing a spatial inductive bias appropriate for the periodic KS domain.

**Training and evaluation.** All methods undergo a grid search over the OT loss weight $\lambda$, and best metrics are reported for each method. Rollouts are evaluated over 1,000 steps on both clean and noisy trajectories. All methods use teacher forcing over sliding windows of length 100 and stride 2.

### D.3. Experimental details for 2D Kolmogorov Flow

Kolmogorov flow is a two-dimensional incompressible Navier–Stokes system driven by a sinusoidal body force, widely used as a benchmark for turbulent dynamics (Chandler & Kerswell, 2013).

**System and dataset.** We consider the 2D Navier–Stokes equations in vorticity form on a periodic domain:

$$\frac{\partial \omega}{\partial t} + D(\omega, \psi) = \frac{1}{Re}\nabla^2\omega - \chi\omega + f \tag{29}$$

$$\nabla^2\psi = -\omega, \tag{30}$$

where the velocity field $\mathbf{v} = (v_x, v_y)$ is recovered from the streamfunction via:

$$v_x = \frac{\partial\psi}{\partial y}, \qquad v_y = -\frac{\partial\psi}{\partial x}. \tag{31}$$

The vorticity is $\omega = (\nabla \times \mathbf{v}) \cdot \hat{z}$, and $D(\omega, \psi) = \frac{\partial\psi}{\partial y}\frac{\partial\omega}{\partial x} - \frac{\partial\psi}{\partial x}\frac{\partial\omega}{\partial y}$ is the system Jacobian. We explore the Kolmogorov flow regime with Reynolds number $Re = 10^4$, time-constant forcing $f$ at a given wavenumber and drag coefficient $\chi = 0.1$.

Observations are corrupted with additive Gaussian noise with standard deviation $0.3 \cdot \mathrm{std}$, where $\mathrm{std}$ denotes the empirical standard deviation of the corresponding clean trajectory. All simulations are performed with the py2d library (Jakhar et al., 2024).

**Emulator architecture.** We use a UNet encoder-decoder with spectral convolution layers, following the architecture proposed in Jiang et al. (2025) for this class of problems.

**Summary and critic architectures.** For the learnable OT variants, the MLP summary network used in the 1D experiments is replaced by a convolutional summary map. This network applies a stack of 2D convolutions with circular padding to each vorticity snapshot, providing a spatial inductive bias appropriate for 2D periodic fields while preserving the same adversarial training framework as in the 1D setting. The critic architecture for the WGAN variant is unchanged from the two previous experiments.

**Training and evaluation.** Training uses teacher forcing, warm-start pretraining on the MSE objective, and subsequent activation of the OT term. Metrics are identical to those reported in the main experiments: L1 histogram error over joint distributions of the NS-equation statistics, radially-averaged spectral distance, and one-step RMSE.

### D.4. Experimental details for L-63

**Experimental Setup.** We generate trajectories from the Lorenz-63 system

$$\dot{x} = \sigma(y - x), \quad \dot{y} = x(\rho - z) - y, \quad \dot{z} = xy - \beta z \tag{32}$$

with a set of parameters sampled around the classic setting $\sigma = 10$, $\rho = 28$, $\beta = 8/3$, producing chaotic dynamics on the characteristic butterfly attractor. Training data consists of 100 crops of 10,000 timesteps each (dt = 0.01), subsampled with a stride of 10 steps and corrupted with the same additive Gaussian $\sigma_{\mathrm{noise}}$ as earlier.

We train a multilayer perceptron emulator $g_\psi : \mathbb{R}^3 \to \mathbb{R}^3$ with four hidden layers, each 128 units and GELU activation function to predict the next states. The emulator is regularized using Sinkhorn divergence ($\varepsilon = 0.05$, $p = 2$) computed on learned summary embeddings. Training uses anchored rollouts similarly to the Lorenz-96 setup, anchoring states every 2 timesteps, computing the OT distance between summary statistics of predicted and true trajectories over these windows

Given the low dimensionality of the Lorenz-63 system and our focus on interpretability, we explore two learnable summary architectures: *Linear summary:* $f_\varphi : \mathbb{R}^3 \to \mathbb{R}^d$ applies a single linear layer, yielding an interpretable one-dimensional projection whose wights $w \in \mathbb{R}^d$ can be directly analyzed. *Nonlinear summary:* $f_\varphi : \mathbb{R}^3 \to \mathbb{R}^d$ uses a three-layer MLP with hidden dimensions [64, 32] and GELU activation functions, allowing the network to discover nonlinear summary statistics.

Training alternates between one step updating the summary network $f_\varphi$ to maximize the Sinkhorn divergence and one step updating the emulator $g_\psi$ to minimize the combined reconstruction and OT loss. We apply gradient clipping with threshold 1.0 to both networks. OT regularization begins after a 10-epoch warm-up period with $\lambda_{\mathrm{OT}} = 0.2$.

# E. Additional Results for Lorenz-96

## E.1. Multi-trajectory Lorenz-96 results

For the L96 multi-trajectory setting, we report median performance with 25th and 75th percentiles to match the evaluation protocol of Jiang et al. (2023).

*Table 8.* Performance on noisy Lorenz–96 trajectories ($\sigma = 0.3$). HS denotes handcrafted fixed summary statistics, while LS denotes learnable summaries. Results report median performance (25th, 75th percentile) over runs.

| Method | $d$ | RMSE ↓ | $L^1$ hist. error ↓ | Spectral dist. ↓ |
|---|---|---|---|---|
| **Baseline** | | | | |
| No OT | - | **0.388** (0.384, 0.395) | 0.348 (0.325, 0.377) | 0.321 (0.312, 0.330) |
| **HS** | | | | |
| Fixed OT | 3 | 0.398 (0.395, 0.403) | 0.175 (0.146, 0.208) | 0.141 (0.135, 0.146) |
| **LS** | | | | |
| Sinkhorn | 3 | 0.406 (0.401, 0.413) | 0.175 (0.156, 0.198) | 0.112 (0.105, 0.118) |
| WGAN | 3 | 0.401 (0.398, 0.406) | **0.149** (0.122, 0.184) | **0.049** (0.041, 0.055) |
| WGAN | 1 | 0.392 (0.390, 0.398) | 0.234 (0.206, 0.262) | 0.205 (0.192, 0.217) |

*Table 9.* Performance on clean Lorenz–96 trajectories ($\sigma = 0.0$). HS denotes handcrafted fixed summary statistics, while LS denotes learnable summaries. Results report median performance (25th, 75th percentile) over runs.

| Method | $d$ | RMSE ↓ | $L^1$ hist. error ↓ | Spectral dist. ↓ |
|---|---|---|---|---|
| **Baseline** | | | | |
| No OT | - | **0.302** (0.297, 0.307) | 0.225 (0.211, 0.240) | 0.294 (0.286, 0.301) |
| **HS** | | | | |
| Fixed OT | 3 | 0.313 (0.307, 0.320) | **0.055** (0.050, 0.061) | 0.119 (0.110, 0.128) |
| **LS** | | | | |
| Sinkhorn | 3 | 0.325 (0.321, 0.330) | 0.109 (0.102, 0.115) | 0.083 (0.076, 0.089) |
| WGAN | 3 | 0.318 (0.311, 0.324) | 0.083 (0.064, 0.099) | **0.074** (0.061, 0.092) |
| WGAN | 1 | 0.306 (0.299, 0.312) | 0.091 (0.72, 0.111) | 0.176 (0.163, 0.187) |

## F. Lorenz-63: Additional Results

Due to the low dimensionality of the Lorenz–63 system, quantitative differences between the no-OT, fixed-OT, and learnable-OT models were small. For this reason, we restrict the Lorenz–63 analysis to visualization-based results that highlight the structure of the learned summary representations and the effect of OT regularization on attractor geometry under noise.

### F.1. Geometric structure of learned summaries

Figure 5 shows the Lorenz–63 attractor colored by the learned summary value $\mathbf{s} = f_\varphi(\mathbf{u})$ across the three canonical projections $(x, y)$, $(x, z)$, and $(y, z)$, alongside the projection induced by the dominant eigenvector of the displacement covariance $C_T$ (Proposition A.1).

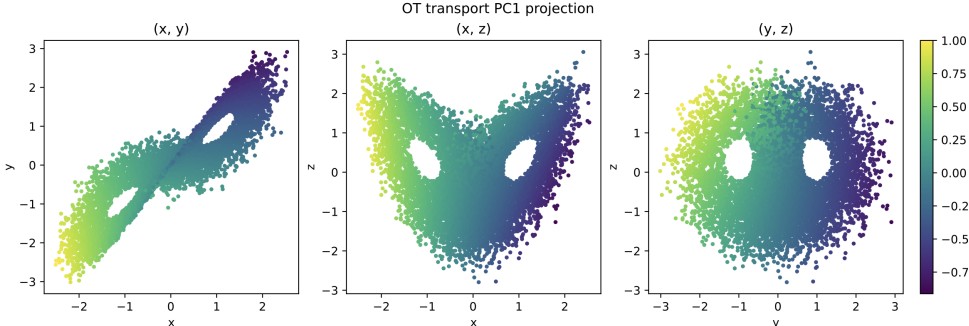

*(a)* Optimal linear summary $f(\mathbf{u}) = \langle \mathbf{v}_{\max}, \mathbf{u} \rangle$, where $\mathbf{v}_{\max}$ is the eigenvector associated with the largest eigenvalue of $C_T$ (Proposition A.1).

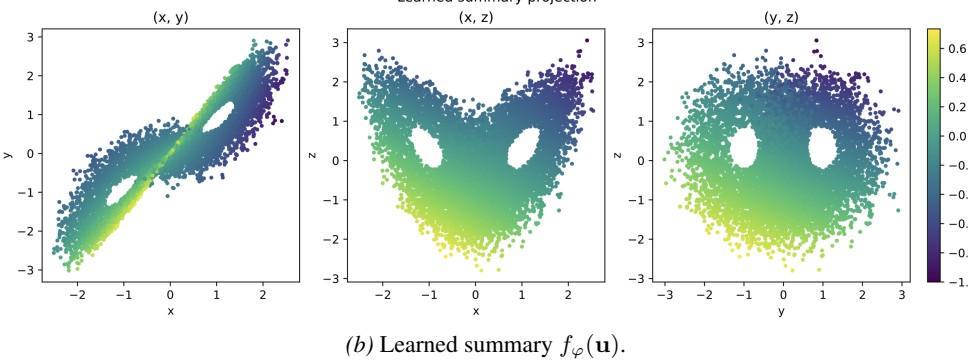

*(b)* Learned summary $f_\varphi(\mathbf{u})$.

*Figure 5.* Lorenz-63 attractor colored by summary value across three orthogonal projections. Color intensity from purple (negative) to yellow (positive) indicates the summary value.

Proposition A.1 identifies linear summary directions that maximize the captured OT discrepancy, with $C_T$ determining the optimal subspace. The visualizations confirm that the learned summaries capture substantial alignment with these transport-optimal directions, supporting the practical relevance of the proposition even when its exact assumptions are not perfectly satisfied.

### F.2. State-space histograms

Figure 6 shows long-run state-space visitation histograms, comparing the ground truth system against the emulator trained with the adversarial OT objective. Both histograms are computed from a single long trajectory after transient removal.

Trajectory Histogram Comparison

*Figure 6.* State-space histograms for Lorenz-63. *Dashed:* Ground truth. *Colored:* Trained emulator.

## F.3. Distribution of learned summary statistics

Figure 7 compares the distribution of the learned summary statistic **s** extracted from ground truth trajectory segments against emulator-generated segments.

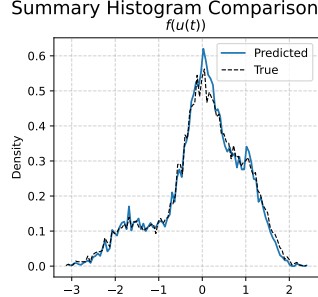

Summary Histogram Comparison

*Figure 7.* Distribution of the learned summary statistic $f_\phi$ for Lorenz-63. *Dashed:* Ground truth trajectories. *Colored:* Emulator trajectories.

# G. Additional Visualization Results

## G.1. L96: Attractor visualization

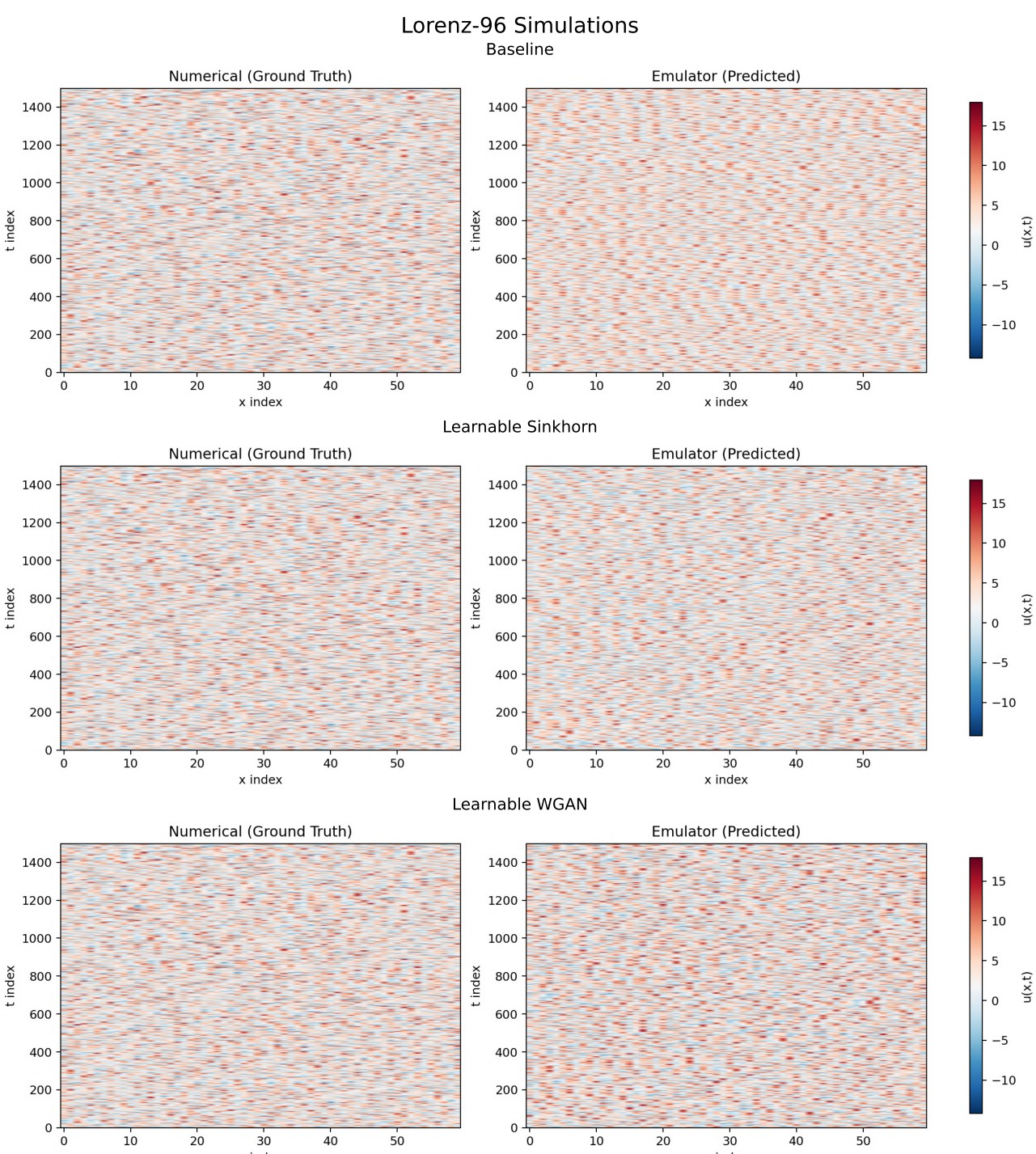

*Figure 8.* Space-time plots of $\mathbf{u}(x, t)$ for the Lorenz–96 system ($d = 60$), comparing ground truth numerical simulations (left) against emulator rollouts (right) over 1,500 timesteps. Each row corresponds to a different method. As expected for chaotic systems, pointwise trajectory agreement is not maintained beyond the Lyapunov time; the relevant comparison is the statistical structure of the attractor over long horizons (c.f. Tables 2.

## G.2. KS: Attractor visualization

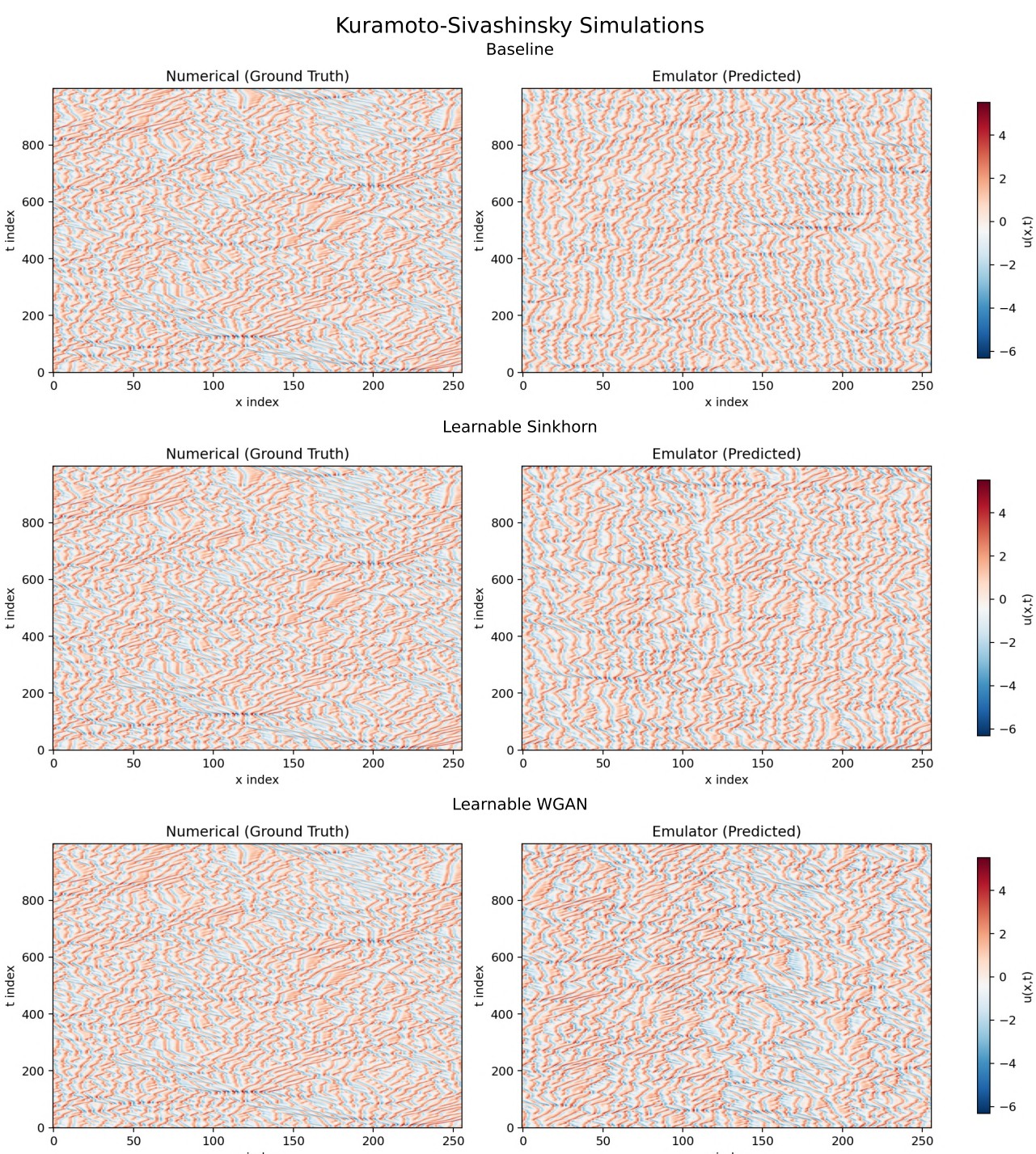

*Figure 9.* Space-time plots of $\mathbf{u}(x, t)$ for the Kuramoto–Sivashinsky equation ($d = 256$), comparing ground truth numerical simulations (left) against emulator rollouts (right) over 1,000 timesteps. Each row corresponds to a different method. Trajectory-level correspondence is not expected beyond the Lyapunov time due to chaos; the relevant comparison is the statistical structure of the attractor rather than pointwise agreement. See Tables 2 for computed metrics.

### G.3. 2D Kolmogorov: Attractor visualization

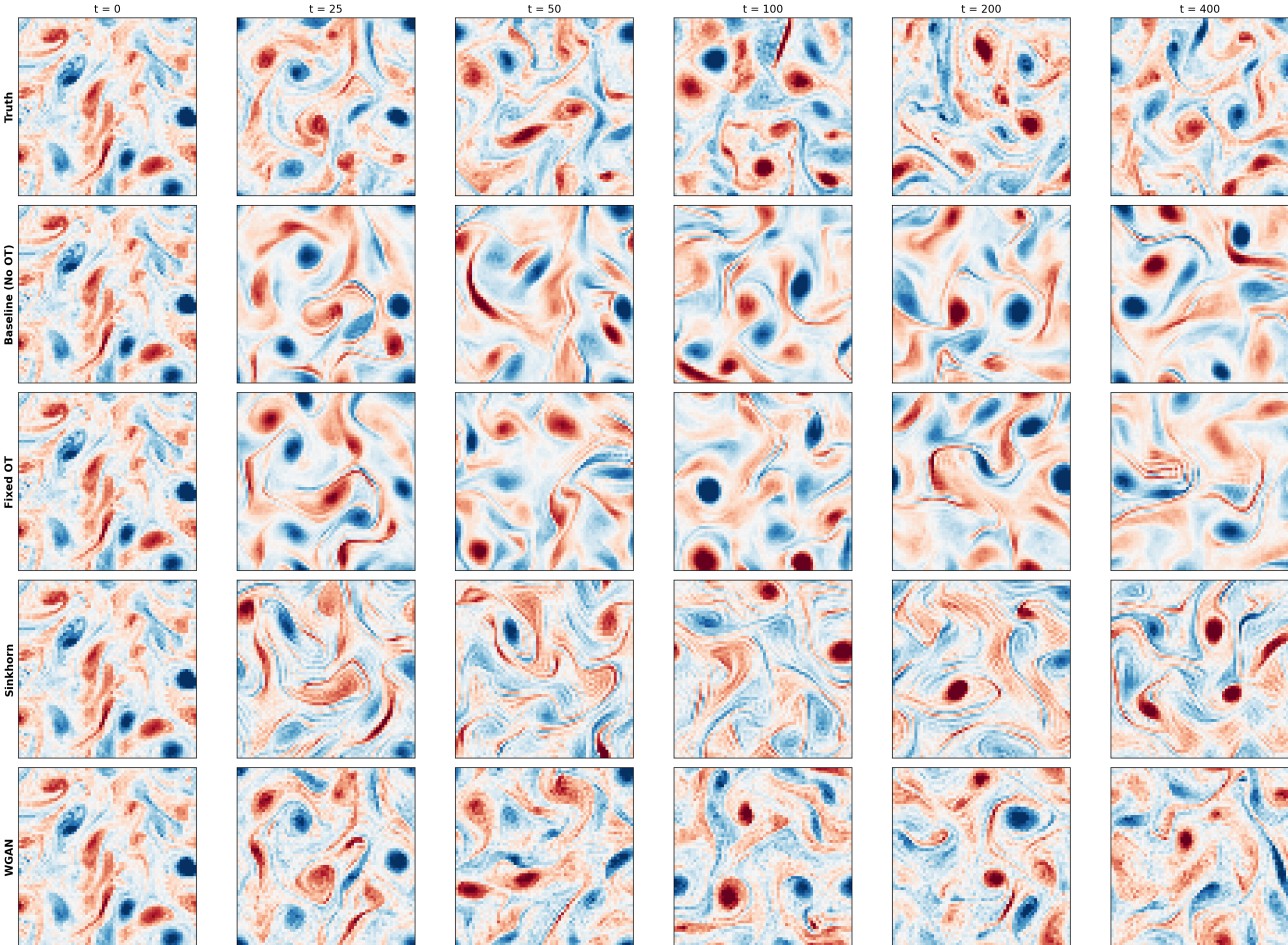

*Figure 10.* Vorticity rollouts for 2D Kolmogorov flow ($\mathrm{Re} = 10^4$, $\alpha = 0.1$) at selected timesteps. Each row shows autoregressive predictions from a different model alongside the ground truth (top row). For quantitative analysis, see Tables 2.

