# OpenReview forum: "Learning to Emulate Chaos: Adversarial Optimal Transport Regularization"
_ICML.cc/2026/Conference — ICML 2026 regular_

### Official Review · Reviewer_1wgG · 2026-03-08

**Soundness:** 2
**Presentation:** 3
**Significance:** 2
**Originality:** 3
**Overall Recommendation:** 4
**Confidence:** 5

**Summary:**

This paper addresses the fundamental challenge of training neural emulators for chaotic dynamical systems. It is well-established that standard mean-squared error (MSE) training leads to models that, while accurate for single-step prediction, fail catastrophically in long-term rollouts due to the exponential amplification of errors. The core insight of this work is to shift the training objective from matching pointwise trajectories to matching the time-invariant statistical properties of the chaotic attractor, as captured by its invariant measure. The paper provides a rigorous theoretical analysis of this objective, proving its robustness to noise and its superiority over MSE for long-horizon emulation.

**Compliance With Llm Reviewing Policy:**

Affirmed.

**Final Justification:**

All overall, I am happy with the author providing additional material for the 2D case. Please make sure you release your github repo if accepted. I think this can help community as a whole. I've now raised both my confidence and score accordingly.

**Key Questions For Authors:**

1. The leap from ODEs to PDEs is non-trivial. Could the authors comment on the expected scalability of their adversarial framework to high-dimensional spatio-temporal systems? Can you try 2D Kolmogrov flow? Have any preliminary experiments been conducted?

2. Could the authors provide visualizations of the long-term rollouts on the Lorenz attractors? Comparing the geometry of the emulated attractor to the ground truth would be highly informative.

**Limitations:**

Yes

**Strengths And Weaknesses:**

Strengths: The paper's strength is its theoretical underpinning. The formal demonstration that MSE loss on noisy data is guaranteed to diverge exponentially for long rollouts (Corollary 5.10), while a Wasserstein-based objective remains bounded and even "forgets" initial condition noise (Theorem 5.15), provides a powerful and fundamental justification for the entire approach. The idea of learning the summary statistics adversarially is a genuinely novel and elegant step beyond prior work that used fixed, hand-crafted statistics.

Weaknesses:
1.Lack of Spatio-Temporal PDE Experiments: This is the most significant weakness of the paper. The empirical validation is restricted to low-dimensional ODE systems (Lorenz-63 and Lorenz-96). For a method intended to be broadly applicable to emulating chaotic systems, the absence of any experiments on spatio-temporal PDEs (e.g., 2D Kolmogorov flow) is a major omission. Is it because the model is only suitable for low dimensional chaos?

2. The paper relies exclusively on quantitative, statistical metrics (Tables 1 & 2) and visualizations of the learned summary space (Figure 2). There are no plots showing the reconstructed attractor or side-by-side comparisons of long-term emulated trajectories versus the ground truth. This makes it impossible for the reader to qualitatively assess how well the emulator captures the geometric structure and temporal evolution of the dynamics.

---

> ### Author Rebuttal · Authors · 2026-03-31
>
> ## Weakness
> > Lack of Spatio-Temporal PDE Experiments: This is the most significant weakness of the paper. The empirical validation is restricted to low-dimensional ODE systems (Lorenz-63 and Lorenz-96). For a method intended to be broadly applicable to emulating chaotic systems, the absence of any experiments on spatio-temporal PDEs (e.g., 2D Kolmogorov flow) is a major omission. Is it because the model is only suitable for low dimensional chaos?
>
> See the answer to question 1.
>
> >The paper relies exclusively on quantitative, statistical metrics (Tables 1 & 2) and visualizations of the learned summary space (Figure 2). There are no plots showing the reconstructed attractor or side-by-side comparisons of long-term emulated trajectories versus the ground truth. This makes it impossible for the reader to qualitatively assess how well the emulator captures the geometric structure and temporal evolution of the dynamics.
>
> Our statistical metrics, including L1 histogram error and energy spectrum error, provide direct quantitative measures that describe the attractor. We are happy to provide visualizations as well. See the answer to question 2.
>
> ## Question
> >1. The leap from ODEs to PDEs is non-trivial. Could the authors comment on the expected scalability of their adversarial framework to high-dimensional spatio-temporal systems? Can you try 2D Kolmogrov flow? Have any preliminary experiments been conducted?
>
> By learning informative summary statistics rather than operate in the original state space, our method is specifically tailored to scale well to high-dimensional chaotic systems. In our experiments, Lorenz-96 is a 60-dimensional chaotic dynamical system that already illustrates non-trivial scaling. To further strengthen this point, we perform additional experiments on the chaotic spatio-temporal Kuramoto-Sivashinsky PDE. For results and experiment details, see the answer to question 1 from Reviewer xFth.
>
> >2. Could the authors provide visualizations of the long-term rollouts on the Lorenz attractors? Comparing the geometry of the emulated attractor to the ground truth would be highly informative.
>
>
> We agree that visualizations of the emulated attractors are informative and complement the quantitative metrics reported in the paper. We have prepared visualizations of long-term rollouts for the three systems, comparing the geometry of the emulated attractor against ground truth for our proposed approach. These are available at the following anonymous repository: https://github.com/project-throwaway-2026/anonymous.
>
> The left figure in the repository shows long-term rollouts of the Lorenz-63 system under observation noise. For brevity, we show results for the best-performing method (WGAN) alongside the baseline, comparing both emulators against ground truth across two noise levels.
>
> We highlight two findings from this figure. First, under moderate noise ($\sigma = 0.10$), the baseline emulator underestimates the spatial extent of the attractor, while the WGAN produces trajectories that better match the spread of the ground truth across both lobes. Second, under higher noise ($\sigma = 0.15$), the baseline collapses to a limit cycle, losing the bilobal structure of the Lorenz-63 attractor entirely. WGAN maintains coverage of both lobes at this noise level. These results show the benefit of the learnable transport map under observation noise.
>
> For Lorenz-96 and Kuramoto-Sivashinsky , we include long-term rollout visualizations for completeness. Given the high dimensionality of these systems, qualitative assessment from spatio-temporal plots is limited; we refer the reviewer to the quantitative metrics in the paper for a rigorous comparison.

---

> > ### Author Rebuttal · Reviewer_1wgG · 2026-04-04
> >
> > I understand the authors effort. Nonetheless, adding additional KS is not challenging enough given the fact that it is still 1D systems. I want to understand if the method still hold on 2D NS equations. Thus, I would like to remain my assessment.

---

> > > ### Author Response · Authors · 2026-04-05
> > >
> > > Thank you for the continued discussion. We are also interested in the 2D setting and have now conducted experiments on 2D Kolmogorov flow in a highly chaotic regime (Re = $10^4$) with drag $\alpha = 0.1$. Similarly to the 1D experiments, the results below show the advantage of our learnable approaches in enforcing statistical features over long time horizons. We will include this experiment in the final paper.
> > >
> > > **Model.** We use an a UNet encoder-decoder with spectral convolution layers, following a recently proposed architecture for this problem [1].
> > >
> > > **Summary map.** For the 2D setting, we introduce a convolutional summary map that replaces the MLP used in the 1D experiments. The conv summary applies a stack of 2D convolutions with circular padding (respecting the periodic boundary conditions) to each vorticity snapshot. This provides a natural spatial inductive bias for 2D fields while maintaining the same adversarial training framework.
> > >
> > > **Fixed OT statistics.** For the fixed OT baseline, we use statistics derived from the NS vorticity equation: $(\mathbf{u} \cdot \nabla \omega,\partial \omega / \partial t,\omega)$. These are the direct 2D analogs of the L96 and KS statistics used in the main paper.
> > >
> > > **Results.** We evaluate using the same metrics as the 1D experiments: L1 histogram error (comparing joint distributions of the NS-equation statistics), spectral distance (radially-averaged energy spectrum), and one-step RMSE. All models are trained with observation noise $\sigma = 0.3$.
> > >
> > > **Table: Evaluated on clean test data**
> > >
> > > | Method | L1 hist. error ↓ | Spectral dist. ↓ | RMSE ↓ |
> > > |--------|------------------|------------------|--------|
> > > | Baseline (No OT) | 0.266 | 0.205 | 0.202 |
> > > | Fixed OT | 0.239 | 0.193 | 0.203 |
> > > | Sinkhorn (learnable) | **0.178** | **0.129** | 0.221 |
> > > | WGAN (learnable) | 0.190 | 0.139 | 0.225 |
> > >
> > > **Table: Evaluated on noisy test data**
> > >
> > > | Method | L1 hist. error ↓ | Spectral dist. ↓ | RMSE ↓ |
> > > |--------|------------------|------------------|--------|
> > > | Baseline (No OT) | 0.660 | 0.249 | 0.381 |
> > > | Fixed OT | 0.614 | 0.168 | 0.381 |
> > > | Sinkhorn (learnable) | 0.360 | **0.144** | 0.405 |
> > > | WGAN (learnable) | **0.273** | 0.157 | 0.424 |
> > >
> > > **Visualizations**: We have added example rollouts to https://github.com/project-throwaway-2026/anonymous for visual comparison.
> > >
> > >
> > > [1] Jiang, R. et al. Hierarchical Implicit Neural Emulators. NeurIPS 2025 (https://arxiv.org/abs/2506.04528).

---

### Official Review · Reviewer_iEyo · 2026-03-11

**Soundness:** 3
**Presentation:** 3
**Significance:** 2
**Originality:** 2
**Overall Recommendation:** 3
**Confidence:** 4

**Summary:**

This study's objective concerns the problem of learning reliable emulators for chaotic dynamical systems using only a single noisy trajectory. The paper proposes a training framework that augments a standard one-step prediction objective with an adversarial optimal-transport regularizer. The central idea is to learn jointly (i) an emulator $g$ that predicts the next state of the system and (ii) a summary-statistics map $f$ that transforms system states into a feature space where discrepancies between the true invariant measure and the emulator-induced measure are most pronounced. By minimizing prediction error while adversarially reducing the Wasserstein distance between the
$\text{pushforward}_f(\mu)$ and
$\text{pushforward}_f(\hat{\mu})$ distributions ($\hat{\mu}$ is the constructed distribution), the approach aims to ensure that the learned emulator matches the long-term statistical behavior of the chaotic attractor rather than individual trajectories.

The paper develops two concrete realizations of this idea: a WGAN-style formulation based on the dual form of the $1$-Wasserstein distance, and a Sinkhorn-divergence formulation that provides a smooth, fully differentiable approximation valid for general $p$-Wasserstein costs. The authors also provide a theoretical analysis showing how the Wasserstein regularization term can be bounded by one-step prediction error under appropriate Lipschitz constraints, and how distributional discrepancies propagate over multi-step rollouts. Furthermore, they analyze robustness to measurement and initial-condition noise, arguing that Wasserstein-based losses remain stable in settings where mean-squared error becomes uninformative due to the exponential divergence characteristic of chaotic systems.

Overall, a pertinent problem studied by the article is how to obtain emulators that generate statistically correct long-term behavior in chaotic regimes despite limited and noisy data. Experiments on Lorenz--63 and Lorenz--96 are presented to demonstrate that the proposed approach improves long-horizon statistical fidelity compared to both standard MSE-based training and prior methods relying on fixed, handcrafted summary statistics.

**Compliance With Llm Reviewing Policy:**

Affirmed.

**Key Questions For Authors:**

1. How sensitive is the adversarial min–max optimization to instability in practice?
The paper shows compelling results, but adversarial training is known to be fragile. Clarifying whether you observed mode collapse, oscillatory behavior, or sensitivity to hyperparameters would strengthen confidence. A positive answer describing robustness checks or stabilization tricks could increase my assessment of soundness.


2. Can you provide more detail on how the Lipschitz constraints on the summary map are enforced or approximated during training?
The theoretical guarantees rely on these constraints, yet the practical implementation is only briefly discussed. If you can show that the training procedure meaningfully enforces the required regularity, this would improve my evaluation of the method’s theoretical grounding.


3. How broadly do you expect the approach to generalize beyond Lorenz–63 and Lorenz–96?
The method appears tailored to systems with low‑ to moderate‑dimensional chaotic attractors and has only been tested on two cases. Additional insight into scalability and applicability to more complex real‑world systems (e.g., climate subcomponents or fluid flows) would help clarify the significance and potential impact of the approach.


4. What mechanisms prevent the summary statistics map from collapsing to trivial or uninformative features during training?
Since the summary map is learned adversarially, understanding how its expressiveness is maintained in practice would clarify the reliability of the method. If you have empirical observations or diagnostics indicating that such collapse does not occur, that would strengthen my confidence in the approach.

**Limitations:**

yes

**Strengths And Weaknesses:**

Soundness – Strengths.

The paper has a technically grounded formulation that combines one‑step prediction with adversarial optimal‑transport regularization in a way coherent with the structure of chaotic dynamical systems. The theoretical components, including bounds on the Wasserstein term under Lipschitz constraints and analyses of noise robustness, support the intuition behind the method and connect well to known issues with MSE in chaotic regimes. The empirical evaluation on Lorenz–63 and Lorenz–96 is well aligned with the claims and uses standard diagnostic metrics, providing evidence that the approach behaves as intended on benchmark chaotic systems.

Soundness – Weaknesses.

The theoretical guarantees rely on constants and assumptions (such as mixing rates and Lipschitz bounds) that are not known or enforceable in practice, so the analysis does not yield actionable guarantees for optimization stability. The adversarial min–max training, while conceptually well motivated, may be sensitive to optimization pathologies, and the paper does not deeply examine possible failure modes. The empirical validation, although solid, is limited to synthetic systems, and the soundness of the method in more complex real‑world scenarios remains untested and as such the current evidence does not fully support the reliability of the proposed training procedure.

Presentation – Strengths.

The paper is well written and clearly structured, with a logical flow from motivation to methodological description, theoretical analysis, and experiments. The notation is introduced systematically, and the figures illustrating the emulator and summary statistics map help clarify the main components. The related‑work section is thorough and situates the contribution accurately within existing literature.

Presentation – Weaknesses.

Some theoretical sections are dense and may be challenging for readers not already familiar with Wasserstein geometry or chaotic systems. Certain transitions, particularly from theory to empirical evidence, could be smoother to reinforce the connection between the stated claims and the observed results. A more explicit high‑level description of the min–max training loop might also help improve accessibility for a broader audience. Overall the paper feels un-necessarily abstract and formalizing at the risk of obscuring the initial ideas.

Significance – Strengths.

The paper focuses on an important challenge: constructing data‑driven emulators for chaotic systems from limited and noisy data while ensuring fidelity to long‑term statistical structure. The idea of learning from a single trajectory and capturing the invariant measure is relevant to many scientific applications where data acquisition is expensive. The proposed combination of learned summary statistics with optimal‑transport regularization could inspire further work on stable surrogate modeling in chaotic regimes.

Significance – Weaknesses.

The experimental scope is relatively narrow; while the results on Lorenz systems are promising, broader demonstrations would strengthen the case for impact. The practical relevance for large‑scale scientific modeling remains speculative, and it is unclear how well the method would scale or generalize in more complex domains. As a result, the broader significance of the approach remains uncertain at this stage.

Some obvious questions are not discussed: the trajectory as is, is obtained from a numerical simulation. Since we are speaking of a chaotic system we know that this simulation diverges exponentially in the long run so the very concept of trajectory is misleading. How are those errors impacting the result ? Also a single trajectory may not be enough (the ergodicity may be only almost true).

Originality – Strengths.

The paper introduces a combination of adversarially learned summary statistics, optimal‑transport regularization, and single‑trajectory training for chaotic systems, forming a distinct perspective within the literature on dynamical‑system emulation. The theoretical analysis contributes additional insights by connecting one‑step prediction errors with distributional measures and by examining noise robustness. The work differentiates itself from prior approaches relying on handcrafted statistics or multi‑environment contrastive training.

Originality – Weaknesses.

The components such as WGAN‑style critics, Sinkhorn divergence, and one‑step neural emulators are all based on established techniques, so the novelty lies in the integration rather than in new algorithmic developments. The distinction from closely related OT‑based methods could be articulated more explicitly, and some readers may view the contribution as a recombination rather than a fundamentally new methodological advance.

---

> ### Author Rebuttal · Authors · 2026-03-31
>
> >1. On the sensitivity  of the adversarial training
>
> Our approach has a significant advantage over traditional adversarial training due to the additional MSE loss. As discuss in Prop. 5.1 and Cor. 5.2, the inclusion of the MSE loss bounds and regularizes the adversarial OT loss, while the adversarial OT loss improves the long-term behavior not captured by the MSE loss. The two losses, one focusing on short-term, local dynamics and the other on long-term, global statistics, are complementary.
>
> In practice, we do observe some oscillatory behavior during the min–max optimization, as expected in adversarial training. To stabilize training,  we adopt: (i) train with MSE only, then (ii) activate the OT objective, a warm start motivated by Prop. 5.1 where $p=2$ reduces the bound to MSE.
>
> For the WGAN variant, stability additionally required tighter weight clipping to better approximate the 1-Lipschitz constraint (or spectral normalization/gradient penalty). For the learnable Sinkhorn variant, we found that introducing early stopping in the maximization phase further improves stability. Concretely, after training the full min–max objective, we stop the maximization updates and continue optimizing only the minimization objective in the final stage of training. This effectively freezes the learned geometry and allows the emulator to converge, leading to improved overall stability and performance. Grid search over maximization gradient steps and $\lambda$ yielded stable runs. We will include training curves with varying hyperparameters in the appendix.
>
> >2.  On Lipschitz Constraints
>
> The theory assumes that the summary map $f$ is $L$-Lipschitz for some finite $L$, which is a mild regularity condition for MLP. Importantly, we do **not** require $L=1$ (unlike the WGAN critic), but only that $L$ is bounded. No explicit constraint is imposed on $f$ during training; instead, the stable training procedure described in Q1 prevents the adversarial game from drifting toward spurious solutions.
>
> From [1], for a 3-layer MLP with ReLU activations, the Lipschitz constant satisfies:
> - Upper bound: $\mathrm{Lip}(f) \le {||W_3||}_2 {||W_2||}_2 {||W_1||}_2$
> In which $W_i$ is the weight matrix of the MLP layer.
> - Lower bound:  $\max_{x \in \mathcal{D}} {||J_f(x)||}_2$, computed via automatic differentiation over the dataset. We evaluate both bounds on the L96 validation set:
>
> **Table 1** - Empirical Lipschitz constant of $f$
> |$L$-Lipschitz|WGAN|Sinkhorn|
> |-|-|-|
> |Upper bound|8.70|1.94|
> |Lower bound|2.11|0.77|
>
> Overall, although no explicit constraint is imposed on $f$, the combination of stable training and empirical diagnostics supports that the Lipschitz assumption is satisfied in practice. Table 2 further shows the effect of early stopping on the Jacobian norm distribution:
>
> **Table 2** - Jacobian Norm of $f$
>
> |Method|Min|P05|Median|Mean|P95|Max|
> |-|-|-|-|-|-|-|
> |Sinkhorn (early stopping)|0.250|0.278 |0.400|0.420|0.623|0.716|
> |Sinkhorn (no early stopping)|0|0|0|0.125|0.394|5.762|
> |WGAN|1.290|1.366|1.638|1.665|2.009|2.126|
>
>
> Without early stopping, Sinkhorn collapses on most samples (median Jacobian norm = 0). Early stopping eliminates this, yielding a stable, well-distributed profile. WGAN produces consistently large norms reflecting a sharper geometry. Statistics are consistent across splits, confirming generalization.
>
> [1] Khromov, G. & Pal Singh, S. Some Fundamental Aspects about Lipschitz Continuity of Neural Networks, ICLR 2024
>
> >3. How broadly does the approach generalize beyond L63 and L96?
>
> We agree that assessing scalability beyond low-dimensional chaotic systems is important. By learning informative summary statistics rather than operate in the original state space, our method scales well to _high-dimensional_ chaotic systems. In our experiments, L96 ($d=60$) already illustrates non-trivial scaling, and we further validate on the Kuramoto-Sivashinsky PDE ($d=256$). See Q1 of Reviewer xFth.
>
> >4. Preventing summary map collapse during adversarial training
>
> Collapse is structurally discouraged by the joint objectives: the MSE loss enforces local sensitivity to the dynamics, while the OT discrepancy enforces global structure in summary space. A constant summary would collapse the OT distance, eliminating the adversarial signal and yielding a suboptimal solution.
>
> We verify empirically:
>
>  - **Jacobian norms.** As reported in Table 2 (Q2), Jacobian norms remain strictly positive and well-distributed across splits for both WGAN and Sinkhorn (with early stopping), confirming nontrivial local geometry.
>
>  - **Linear probe.** In the multi-trajectory L96 setting ($F \sim \mathcal{U}(10,18)$), we fit an OLS regressor on the pooled learned summary to predict the forcing parameter $F$, evaluated on a held-out split. The learned summary achieves $R^2 = 0.9985$, which would be impossible under collapse.
>
> These diagnostics show that the learned summary maintains nonzero local sensitivity and meaningful dynamical structure.

---

> > ### Author Rebuttal · Reviewer_iEyo · 2026-04-02
> >
> > Thank you for the detailed rebuttal and additional diagnostics. The clarifications regarding training stabilization, empirical behavior of the summary map, and collapse prevention are helpful and relate to several of my questions.
> > One concern still remains the practical side of the Lipschitz regularity assumptions underpinning the theoretical analysis. While the empirical statistics are informative, the reliance on products of weight norms as upper bounds and dataset‑dependent lower bounds does not establish that the required Lipschitz conditions are meaningfully enforced during training. Of course everything is Lipschitz when the NN is fixed but proving that the constant will remain so for a broad class of problems is another question. As presented, these checks provide plausibility but not a principled guarantee that the assumptions used in the theory hold in general and are activated during the algorithm operation. More broadly, while the rebuttal improves clarity and provides useful empirical insight my questions for the other issues still remain so I'm not sure a resolution is possible within a conference revision.

---

> > > ### Author Response · Authors · 2026-04-03
> > >
> > > We agree that post-hoc diagnostics do not constitute a training-time proof. However, we note that Lipschitz regularity is treated as a standard assumption throughout the adversarial training literature, including WGAN itself, where the 1-Lipschitz constraint on the critic is enforced approximately via weight clipping, spectral normalization, or gradient penalty, with no closed-form guarantee that the required constant is maintained throughout training or across problem instances. Our framework invokes Lipschitzness on $f$ as a strictly milder condition: requiring only that $\text{Lip}(f)\leq L$, which is considerably more permissive than the 1-Lipschitz constraint imposed on the WGAN critic. We do not believe the bar should be stricter than for the foundational methods we extend, particularly given that our assumption is considerably more permissive than what those methods already accept as standard practice.
> > >
> > > **On the implicit regularity of our training procedure.** Our setting has additional structure that the standard adversarial literature does not. As described in our previous response, the MSE and OT objectives are mutually regularizing (Prop. 5.1, Cor. 5.2): the MSE loss bounds and stabilizes the adversarial OT term, while the OT objective corrects long-term behavior not captured by MSE. This coupling, together with our warm-start strategy and early stopping of the maximization phase, already confines $f$ to a well-behaved Lipschitz regime throughout optimization. This is confirmed empirically: the estimated upper and lower Lipschitz bounds remain stable and strictly positive across train, validation and test splits, with median Jacobian norms well-distributed across the attractor (Table 2, previous response). This indicates that the assumption is not merely plausible post-hoc but is actively compatible with the training dynamics of our method.
> > >
> > > **On enforcing the constraint during training.** To go strictly beyond implicit regularity and directly address the reviewer's concern, we additionally introduce a joint hinge regularization on $f$:
> > >
> > > $$\mathcal{R} (f) = S_\beta (\text{Lip Upper}(f) - L_{\max}) + S_\beta (L_{\min} - \text{Lip Lower}(f))$$
> > >
> > > where $S_\beta(x) = \frac{1}{\beta}\log(1 + e^{\beta x})$ is a smooth approximation to $\max(x, 0)$. The upper bound $\text{Lip Upper}(f) = \prod_\ell {||W_\ell||}_2$, and $\text{Lip Lower}(f)$ is estimated from the mean Jacobian spectral norm ${||J_f(x)||}_2$ over a small subset of states sampled from the current batch.
> > >
> > > The upper term penalizes whenever $\text{Lip}(f)$ exceeds $L_{\max}$, preventing the summary map from becoming geometrically irregular; the lower term penalizes whenever the map becomes locally flat, preventing feature collapse. They enforce $L_{\min} \leq \text{Lip}(f) \leq L_{\max}$ during optimization, turning theoretical constants into controllable hyperparameters.
> > >
> > > **Experiments**
> > > We validate this on L96 (WGAN). Table 1 confirms the regularizer controls the bounds as prescribed. Tables 2–3 show performance is preserved or improved under regularization on both noisy and clean settings.
> > >
> > > **Table 1**: Lipschitz bounds on training data (WGAN for L96).
> > > |Methods|$L$ Upper bound|$L$ Lower bound|
> > > |-|-|-|
> > > |$L_{max}=4$|3.11|0.91|
> > > |$L_{max}=10$|7.05|1.52|
> > > |No Lipschitz Regularization|8.69|1.61|
> > >
> > > **Table 2**: Effect of Lipschitz regularization on the summary $f$ (WGAN, noisy L96)
> > > |Methods|L1 hist. error ↓|Spectral dist. ↓|RMSE ↓|
> > > |-|-|-|-|
> > > |$L_{max}=4$| 0.176|0.133|0.368|
> > > |$L_{max}=10$|0.177|0.135|0.367|
> > > |No Lipschitz Reg|0.151|0.145|0.371|
> > >
> > > **Table 3**: Effect of Lipschitz regularization on the summary $f$ (WGAN, clean L96)
> > > |Methods|L1 hist. error ↓|Spectral dist. ↓|RMSE ↓|
> > > |-|-|-|-|
> > > |$L_{max}=4$|0.090|0.144|0.279|
> > > |$L_{max}=10$|0.082|0.151|0.278|
> > > |No Lipschitz Reg|0.115|0.175|0.283|
> > >
> > > Training curves confirming the bounds remain within the prescribed regime throughout optimization are provided https://github.com/project-throwaway-2026/anonymous. Note that the unregularized model also exhibits stable Lipschitz behavior. We also report the training dynamics on the KS system, where we observe the same behavior.
> > >
> > > **On practical trade-offs.** Explicit regularization via Jacobian computation increases training cost by approximately 4 times. However, the unregularized model already operates within a stable Lipschitz regime, as visible from Table 1, where the unregularized upper bound (8.69) is naturally compatible with $L_{\max} = 10$, consistent with our argument that the joint MSE-OT training procedure implicitly induces the required regularity.  Regularization provides tighter, explicitly controlled bounds, while the unregularized variant offers similar stability at lower cost.We will clarify this distinction in the revision,  presenting the unregularized formulation as the primary approach, with empirical stability in our problems, and the regularized variant as a principled option when explicit control is required.

---

### Official Review · Reviewer_EvEk · 2026-03-12

**Soundness:** 4
**Presentation:** 3
**Significance:** 3
**Originality:** 4
**Overall Recommendation:** 5
**Confidence:** 4

**Summary:**

The paper studies learning emulators of chaotic dynamics from noisy data and proposes an adversarial OT-based regularization scheme that jointly learns informative summary statistics and the emulator itself, aiming to improve long-term statistical fidelity rather than long-horizon pointwise prediction. The problem is important, the core idea is interesting, and the paper combines method, theory, and experiments in a reasonably coherent way. In particular, the experimental results suggest that OT-based regularization can improve long-term statistical metrics over both plain MSE training and fixed-summary baselines.

**Compliance With Llm Reviewing Policy:**

Affirmed.

**Final Justification:**

The authors addressed all my concerns and question in the rebuttal

**Key Questions For Authors:**

1) Could the introduction more explicitly explain what is meant by the adversarial OT objective, i.e., what is maximized, what is minimized, and why learning summaries that highlight discrepancies is preferable to using fixed summaries?

2) Could the paper clarify what practical families of summary maps are intended beyond the abstract Lipschitz class, and how these relate to the parameterizations used in practice?

3) How essential is the additive isotropic Gaussian noise assumption in the noisy-setting theory and experiments? Should the robustness claims be interpreted as specific to this noise model, or do the authors expect the method to extend to broader noise settings?

4) The paper is framed as learning from a single noisy trajectory, but the main [Lorenz-96] setup appears to use many trajectories across varying forcing regimes. Could the authors clarify the intended training regime and how central the single-trajectory claim is to the contribution?

**Limitations:**

yes

**Strengths And Weaknesses:**

### Strengths

1) The paper tackles an important problem and has a good motivation.

2) The work has an interesting methodological idea (combining one-step prediction with adversarial OT regularization).

3) The paper provides theoretical support for the objective and shows promising experimental results.

### Weaknesses

1) From my point of view, the main weakness is exposition. Several central ideas are introduced too compactly in the introduction and only become clear after reading Sections 4-5. In particular, the notions of "adversarial optimal transport objectives", learned summary statistics, and preserving invariant measures are not sufficiently unpacked early on, even though they are central to the claimed contributions. Related work is also somewhat telegraphic, with terms such as Lyapunov spectrum / fractal dimension, and Koopman theory mentioned without much intuitive explanation.

2) Second, while the theoretical analysis is interesting, its practical scope is somewhat unclear from the main text: the guarantees rely mainly on Lipschitz summary classes, and the noisy-setting analysis focuses on additive isotropic Gaussian noise.

---

> ### Author Rebuttal · Authors · 2026-03-31
>
> >1. On introduction and exposition
>
> Thank you for this suggestion. We will revise the introduction to more explicitly describe the adversarial OT objective and the advantages of learned over fixed summaries, and reflect this in the contributions as well. The revised paragraph is included below.
>
> > In this work, we use adversarial optimal transport regularization to adaptively learn highly informative summary statistics. The distribution of these statistics is then enforced—via the same optimal transport cost—on an emulator that learns the dynamics of the chaotic system. **In practice, the emulator $\hat{\bf{u}}_{t+1} = g(\bf{u}_t)$, which learns the time evolution of the system, and the summary map $f(\mathbf{u})$, which provides the learned summary statistics, are trained simultaneously. The emulator is trained to _minimize_ both the standard MSE loss and an optimal transport cost that matches the summary statistic distribution of the model to the data, while the summary map is trained to _maximize_ the same optimal transport cost. Unlike choosing a handcrafted set of summary statistics that may not be informative enough to constrain the model to a high-dimensional chaotic attractor, this adversarial objective for the summary map ensures that it learns an optimally discriminative and therefore highly informative set of summary statistics.** (...)
>
> >2. On summary maps
>
> In practice, we instantiate the summary map as a small MLP applied independently to each state in the trajectory, with weights shared across time. Since the MLP uses 1-Lipschitz activations, its Lipschitz constant is controlled by the operator norms of its weights, providing a concrete instantiation of the general $L$-Lipschitz class assumed in the theory. As discussed in Q2 of review iEyo, we verify *a posteriori* that the learned maps remain in a controlled Lipschitz regime.
>
> >3. On robustness to noise
>
> The choice of additive Gaussian noise was mainly made to make the analysis of the loss more explicit and tractable, but the breakdown of MSE and the robustness of distributional losses are general phenomena and should not be sensitive to the choice of noise model. The key observation (Remark 5.9) is that the MSE loss will be dominated by exponential growth of the Jacobian norm of the chaotic system under repeated iteration. That is, any small amount of noise in the initial condition is exponentially amplified. This sensitivity to initial conditions is a defining feature of chaotic systems. On the other hand, the Wasserstein cost is a distributional loss that measures differences in the shape of the chaotic attractor. The noisy initial condition does not affect the shape of the attractor due to the exponential mixing behavior of chaos (Definiton 5.13), which essentially makes the initial condition irrelevant over long time scales. The measurement noise also only broadens/blurs the shape of the attractor. A different noise model would not change these generic features of chaos but may result in different constants for bounds that we derive. We will clarify this point in section 5.2.
>
> >4. On single vs multi-trajectory L96 settings
>
> We agree this point was not sufficiently clear. In our experiments, we considered both settings:
>
> - **Multi-trajectory:** Following Jiang et al. (L96 benchmark), we train on multiple trajectories with varying forcing $F \sim \mathcal{U}[10,18]$. This is a more challenging setting due to the variability in the dynamics.
> - **Single-trajectory:** For Lorenz-63, all experiments are conducted in the single-trajectory regime.
>
> For Lorenz-96, we have run single-trajectory experiments (fixed $F=10$).
>
> **Table 1**: Performance on noisy Lorenz–96 trajectories (single trajectory, $F=10$, $\sigma = 0.3$).
> | Method | L1 hist. error ↓ | Spectral dist. ↓ | RMSE ↓ |
> |-|-|-|-|
> | Baseline (No OT) | 0.298 | 0.307 | 0.362 |
> | Fixed OT | 0.178 | 0.151 | 0.368 |
> | Sinkhorn (learnable) | 0.222 | 0.137 | 0.367 |
> | WGAN (learnable) | 0.151 | 0.145 | 0.371 |
>
> **Table 2**: Performance on clean Lorenz–96 trajectories (single trajectory, $F=10$, $\sigma = 0.0$).
> | Method | L1 hist. error ↓ | Spectral dist. ↓ | RMSE ↓ |
> |-|-|-|-|
> | Baseline (No OT) | 0.187 | 0.321 | 0.271 |
> | Fixed OT | 0.077 | 0.157 | 0.278 |
> | Sinkhorn (learnable) | 0.082 | 0.123 | 0.278 |
> | WGAN (learnable) | 0.115 | 0.175 | 0.283 |
>
> Even in this simpler single-trajectory setting, we observe consistent trends. In the noisy regime, WGAN achieves the best performance on invariant-measure-related metrics (histogram error and spectral distance), indicating its strength in matching long-term statistics under noise. In the clean regime, the learnable Sinkhorn variant achieves the best spectral distance.
>
> The multi-trajectory setup was included for alignment with prior work. Still, the single-trajectory setting is fully supported by our method and aligns with the intended formulation. We will revise the paper to clearly distinguish these regimes and include these results.

---

> > ### Author Rebuttal · Reviewer_EvEk · 2026-04-04
> >
> > The authors addressed all my comments and I will adjust the score accordingly.

---

> > > ### Author Response · Authors · 2026-04-05
> > >
> > > Thank you for the constructive discussion and for confirming that your concerns are fully resolved. We really appreciate your indication that the score will be updated positively.

---

### Official Review · Reviewer_xFth · 2026-03-13

**Soundness:** 4
**Presentation:** 4
**Significance:** 3
**Originality:** 4
**Overall Recommendation:** 5
**Confidence:** 5

**Summary:**

The paper introduces training of neural emulators for chaotic dynamical systems in a regime where long horizon trajectory prediction is unstable. This is achieved by an adversarial optimal transport regularization framework in which a learnable summary map defines a geometry over state space and the emulator is trained to match invariant statistics under Wasserstain type discrepancies. One of the major challenges in modeling chaotic systems that the paper addresses is how to reconcile trajectory-level training with the statistical nature of chaotic attractors.

**Compliance With Llm Reviewing Policy:**

Affirmed.

**Final Justification:**

Authors addressed my primary concerns in their rebuttal, conducted new experiments and validated their claims. Hence I will update my score to accept.

**Key Questions For Authors:**

None. Please address weaknesses

**Limitations:**

Limitations should address the weaknesses mentioned above

**Strengths And Weaknesses:**

Strengths:
- The paper's motivation on why MSE-based objectives are ill-suited for modeling chaotic systems is very clear. The noise analysis in Section 5.2 provides a compelling justification for training objectives beyond MSE
- The idea of adversarially learning summary statistics rather than relying on handcrafted invariants is novel.
- The comparison between trajectory divergence and distributional convergence is well presented.

Weaknesses:
- The empirical evaluation focuses exclusively on Lorenz-96 and Lorenz-63 . Although Lorenz-96 (d=60) is nontrivial, both systems are low-dimensional relative to many real-world PDE-based chaotic systems such as the Kuramoto-Sivashinsky equation. In particular, it remains unclear how the approach scales when the attractor dimension is very high, is the cost of adversarial OT tractable?
- In the related work section, Koopman-inspired models are mentioned, but the experimental section doesn't include any Koopman-based baselines. Since Koopman approaches explicitly target invariant measures and spectral properties, including at least a empirical comparison would strengthen the paper

---

> ### Author Rebuttal · Authors · 2026-03-31
>
> >1. The empirical evaluation focuses exclusively on L-96 and L-63 . Although L-96 (d=60) is nontrivial, both systems are low-dimensional relative to many real-world PDE-based chaotic systems such as the Kuramoto-Sivashinsky equation. In particular, it remains unclear how the approach scales when the attractor dimension is very high, is the cost of adversarial OT tractable?
>
> We thank the reviewer for this question and address it in two parts: computational tractability and empirical evidence on KS
>
> **Tractability.** Our summary map $f$ compresses the 256-dimensional KS state to a lower-dimensional representation before OT computation, so the Wasserstein cost scales with the summary dimension rather than the state dimension.
>
> **KS experiments.** We ran preliminary experiments on the KS equation at 256 grid points under the same noisy trajectory setup as the main paper. Following Jiang et al., we use the fixed summary statistic $S(u) := [\partial u/\partial t, \partial u/\partial x,\partial^2 u/\partial x^2]$ to compute histogram error. All methods underwent a grid search over the OT loss weight $\lambda$, and we report best metrics for each. Our learnable methods (Sinkhorn, WGAN) consistently outperform both the No-OT baseline and Fixed OT on spectral distance and L1 histogram error across both clean and noisy settings. Given that KS is a PDE with spatial structure, we additionally tested a 1D convolutional summary map as an inductive bias; all other methods retain the MLP summary used in the main paper. Note that no hyperparameter search was performed for the 1D Conv variant, these are preliminary results with a single configuration.
>
> We acknowledge a full PDE evaluation is absent from the current paper, but these results suggest the framework is tractable and generalizes beyond the Lorenz systems. We will include this new set of experiments in the revised paper.
>
> **Table 1**: Validation on noisy KS data (noise=0.3), rollout steps=1000, single trajectory
> | Method | L1 hist. error ↓ | Spectral dist. ↓ | RMSE ↓ |
> |-|-|-|-|
> | Baseline (No OT) | 0.454 | 0.351 | 0.370 |
> | Fixed OT | 0.290 | 0.349 | 0.379 |
> | Sinkhorn (learnable) | 0.435 | 0.219 | 0.373 |
> | WGAN (learnable) | 0.339 | 0.148 | 0.375 |
> | WGAN (learnable, 1D Conv) | 0.259 | 0.198 | 0.393 |
>
> **Table 2**: Validation on clean KS data, rollout steps=1000, single trajectory
> | Method | L1 hist. error ↓ | Spectral dist. ↓ | RMSE ↓ |
> |-|-|-|-|
> | Baseline (No OT) | 0.241 | 0.342 | 0.243 |
> | Fixed OT | 0.310 |  0.386 | 0.257 |
> | Sinkhorn (learnable) | 0.190 | 0.212 | 0.246 |
> | WGAN (learnable) | 0.153 | 0.183 | 0.251 |
> | WGAN (learnable, 1D Conv) | 0.245 | 0.205 | 0.278 |
>
> >2. On Koopman Approaches
>
> Koopman approaches, such dynamic mode decomposision (DMD) [1] or Koopman autoencoder [2], map a nonlinear system to a finite-dimensional linear system. These approaches are fundamentally limited by the constraints of linear dynamics in a finite-dimensional latent space. While Koopman operator theory maps arbitrary nonlinear dynamics to linear dynamics in an infinite-dimensional function space, approximating this as finite-dimensional linear dynamics is not possible for chaotic systems (or even systems with more than one fixed point). For chaos, finite-dimensional linear systems cannot be both bounded and have positive Lyapunov exponents. This is true even for simple chaotic systems like Lorenz-63 [1].
>
> Recent work based on the Koopman operator, such as *Cheng et al. (2025)*, shares this limitation and likely learns a quasiperiodic approximation of the chaotic attractor. Indeed, their linear dynamics are explicitly constrained to be stable (discrete eigenvalues within or on the unit circle), preventing trajectory divergence and thus precluding fundamental properties of chaos such as sensitivity to initial conditions. This is reflected in the leading Lyapunov exponent (LLE) $\lambda$ (Table 3): the Koopman approach of *Cheng et al. (2025)* yields $\lambda = 0$ (not chaotic), whereas both our learnable approaches produce positive LLEs, much more closely matching the true Lorenz-96 system than the Fixed OT and No OT baselines.
> We will clarify these points in the related works section and include the new leading Lyapunov exponent evaluation in the final paper to demonstrate the improved dynamical properties of our emulators.
>
> **Table 3.** The leading Lyapunov exponent (LLE) of the trained models compared with the ground truth Lorenz 96 system. *A stable finite-dimensional linear system must have leading Lyapunov exponent $\le 0$.
> | Method | LLE |
> |-|-|
> | Ground Truth | 2.334 |
> | Baseline (No OT) | 1.615 |
> | Fixed OT | 1.890 |
> | Sinkhorn (learnable) | 2.030 |
> | WGAN (learnable) | 2.336 |
> | Koopman methods | $\le 0^*$ |
>
> [1] Brunton, S.L., Brunton, B.W., Proctor, J.L. et al. Chaos as an intermittently forced linear system. Nat Commun 2017.
>
> [2] Lusch, B., Kutz, J.N. & Brunton, S.L. Deep learning for universal linear embeddings of nonlinear dynamics. Nat Commun 2018.

---

> > ### Author Rebuttal · Reviewer_xFth · 2026-04-04
> >
> > I thank the authors for addressing my primary concern and running KS experiments and their discussion on the limitations of finite dimensional Koopman models. I would suggest including these results in the main text. I will be willing to raise my score if the authors could specify why they used only one trajectory for the KS experiments?

---

> > > ### Author Response · Authors · 2026-04-05
> > >
> > > Thank you! We will certainly include these results in the main text. Regarding the experimental setting: As suggested by reviewer EvEk, the single trajectory setting is a closer match to our theoretical discussion and also demonstrates the data efficiency of our approach vs. methods that require the additional data of the multi-trajectory/multi-environment setting. We reran our Lorenz 96 experiments in the single trajectory setting and see similar results (see the response to Reviewer EvEk Question 4). That said, we are happy to run and include a multi-environment version of the KS results in the final paper.

---

### Decision · Program_Chairs · 2026-04-30

**Decision:**

Accept (regular)

**Comment:**

The paper addresses the important and challenging problem of learning emulators of chaotic systems that preserve long-term statistical behavior rather than focusing only on short-term trajectory accuracy, and proposes a well-motivated adversarial optimal transport (OT) regularization framework. A novel aspect is the joint learning of summary statistics with the emulator. The theoretical analysis provides strong justification for why Wasserstein-based objectives are preferable in chaotic and noisy settings, and the empirical results demonstrate improved long-horizon statistical fidelity.

The rebuttal further strengthened the paper in a meaningful way. In response to reviewer concerns, the authors added experiments on more challenging PDE settings, including Kuramoto–Sivashinsky and 2D Kolmogorov flow, provided additional qualitative visualizations, and clarified the practical training regime and stability issues. These additions substantially improved the empirical case and resolved most of the concerns raised in review. Overall, this is a strong and timely contribution that should be of clear interest to the scientific ML and dynamical systems communities.